# Self-serving biases shape the relationship between future thinking and remembering of elections
Marius Boeltzig [1] ✉, Ricarda I. Schubotz [1], Scott Cole [2,4] & Clare J. Rathbone [3,4]

While there is a strong relationship between remembering and future thinking, it has been unclear whether this persists when constraining participants to one specific significant public event. We employed a unique longitudinal approach to uncover how the differences and similarities between remembering and imagining are influenced by self-serving biases evoked by the event itself. Across three longitudinal questionnaire studies testing participants before and after 2024 elections in Germany ($N = 136$), the UK ($N = 89$), and the USA ($N = 243$), we found evidence for self-serving biases in the congruence between future thinking and remembering. Election winners robustly remembered the election as more important and more vivid than they had imagined it before. In the US study, the inconsistency in attitudes across time caused by this shift was resolved by also misremembering the prediction given before the election, with Harris voters thinking they had predicted a less fair, and Trump voters thinking they had predicted a fairer election than they actually had. Additionally, there was an overestimation of pre-election optimism among Harris voters, possibly to help explain current feelings about the outcome, and an underestimation of optimism for Trump voters, making the win more significant. The results reveal that phenomenological differences between remembering and future thinking are contingent on self-serving biases and indicate that participants misremember previous future thoughts in accordance with current needs and attitudes. These mechanisms can lead to entrenched polarization, as partisan beliefs are reinforced by biased future thinking and remembering.

As anybody who has followed the political tides of 2024 would likely agree, elections can be transformative events. As such, they are memory catalysers: People often form memories about the circumstances of learning about election outcomes[1,2], they can become collectively shared memories[3], and they can propel new leaders that – for better or worse – will be remembered by generations to come[4–7]. It is therefore natural that we do not only remember elections – but also anticipate them, speculate about their results, and imagine how their outcomes might shape our lives[8–10].

However, memories are not always aligned with an objective reality. Instead, they are influenced by a wealth of biases that serve to maintain a positive image of ourselves and our groups[11]. The process of thinking about our future is highly dependent on the process of thinking about our past, at both the personal[12,13] and collective level[14,15]. Nonetheless, there are also well-established past-future differences[15,16]. In three longitudinal studies, involving three different nation-wide elections, we tested whether these differences depend on the outcome of the event itself, and thereby on self-serving biases, adjusting both past and future in a way that boosts self-image. Additionally, we tested whether the incongruencies between future thinking and remembering that arise through these biases can be harmonized by changing memories of future simulations. Answering these hitherto unaddressed questions may elucidate how cognitive biases can contribute to partisan alignments in memory and future thinking, also on a collective level[15,17], and thereby contribute to understanding key socio-cultural issues such as political polarization[18].

Remembering an event involves an attempt to recapture the objective details of a prior happening that sometimes leads to inaccuracies[13]. Future thinking is the capacity to imagine future scenarios, which may or may not come to pass[13]. The ability to mentally travel in time, both into the past and

[1]Department of Psychology, University of Münster, Münster, Germany. [2]School of Education, Language and Psychology, York St John University, York, UK. [3]Centre for Psychological Research, Oxford Brookes University, Oxford, UK. [4]These authors contributed equally: Scott Cole, Clare J. Rathbone. ✉e-mail: marius.boeltzig@uni-muenster.de

future, enables vital cognitive processes such as navigation, decision making, and attitude formation[19–22]. These two directions of mental time travel, remembering and future thinking, are highly related. According to the constructive episodic simulation hypothesis, we flexibly and constructively piece together parts of different memories when we imagine future scenarios[23,24]. Consequently, the two processes overlap in neural mechanisms[13,25] and share properties such as content[12]. Although much less is known about the neural correlates of collective mental time travel, emerging evidence supports the idea that collective future thinking similarly relies on collective memory[15].

However, there are also important phenomenological differences between remembering and future thinking. A meta-analysis found that memories were generally rated as more vividly experienced than future thoughts and that future events were rated as more positive than past events[26]. This future positivity bias has been suggested to be produced by unconstrained positive illusions when thinking about an unknown future, while memories are more tied to the reality of the past[27]. Furthermore, future simulations are usually rated as more important[16] or consequential[27] than past events, perhaps as the human mind is organised around prospection rather than driven by the past[28]. At the collective level, reviews have concluded that collective future thinking is more negative, and less specific, than collective remembering[15,29].

However, previous studies have usually not constrained the target of future thinking and remembering. Some studies used neutral cue words to which participants retrieve a memory or simulate a future event[16], others specified a time frame[30,31], or a valence category[27,29,32]. Consequently, participants are not constrained in which specific events they select for past and future – and the phenomenological differences may consequently arise due to differences in sampling of events instead of differences in the processes themselves (see Fig. 1A). Even when specific collective future events were supposed to be imagined, these were not events that came to pass and would later be remembered for a direct comparison[17].

To test whether differences in vividness, valence, and importance hold even when using *one and the same event* for both processes, it is necessary to ask participants to imagine a specific event before it happens, and to remember it after it happens (see Fig. 1B). There is some precedence for such an approach from the affect forecasting and remembering literature[33] and

work on perinatal mental health[34]. Constraining the event allows for control of temporal distance, thus also eliminating effects of different time frames that could arise in the cue word method[35] and affect phenomenology, which can only be statistically controlled post-study. Our study controls for these factors within its design.

How we remember or imagine future events can be strongly impacted by several systematic biases. People can engage in collective future thinking and remembering, thus mentally traveling in time as a member of a group[14]. Additionally, group memberships can become strongly tied to the self, resulting in a feeling of oneness with a group referred to as identity fusion[36]. This is a process that has been observed for political identities[37,38]. Specifically, the amount of identity fusion increases before elections[39], which consequently become an arena for the negotiation of identities[40–43]. Consequently, it can be assumed that, especially in the context of political elections, wins of one's group (i.e., party) can be seen as one's own wins, and election defeats can become personal defeats. The process under study here is therefore situated directly at the intersection between personal and collective future thinking.

It is therefore productive to draw on three broad classes of self-serving memory biases, distinguished by Schacter and colleagues[11] and apply to them to the intersection between personal and collective. First, the *positivity bias* encompasses a general overestimation of how positive an event was or will be. Second, a *consistency bias* refers to people aligning past events with current attitudes or knowledge. Third, *self-enhancing biases* encapsulate the use of future or past events to enhance one's current appraisal of self.

These biases have been observed to impact both future thinking and memory. Constructions of future events are often positive to boost self-appraisals[32,44], self-congruent as they reflect current concerns and goals[45], and self-enhancing in their high rated importance[16,27]. Especially in the context of elections, participants tend toward wishful thinking with overly optimistic forecasts[8,9].

Memories, albeit generally less positive than future simulations[26], are remembered as more positive than the corresponding event really was[46,47], are strongly influenced by current attitudes[48–50], and more favourable to ourselves than what matches reality[51]. Consequently, these biases also impact remembering in a political context. For instance, whether participants falsely remember fabricated news stories about a politician or the

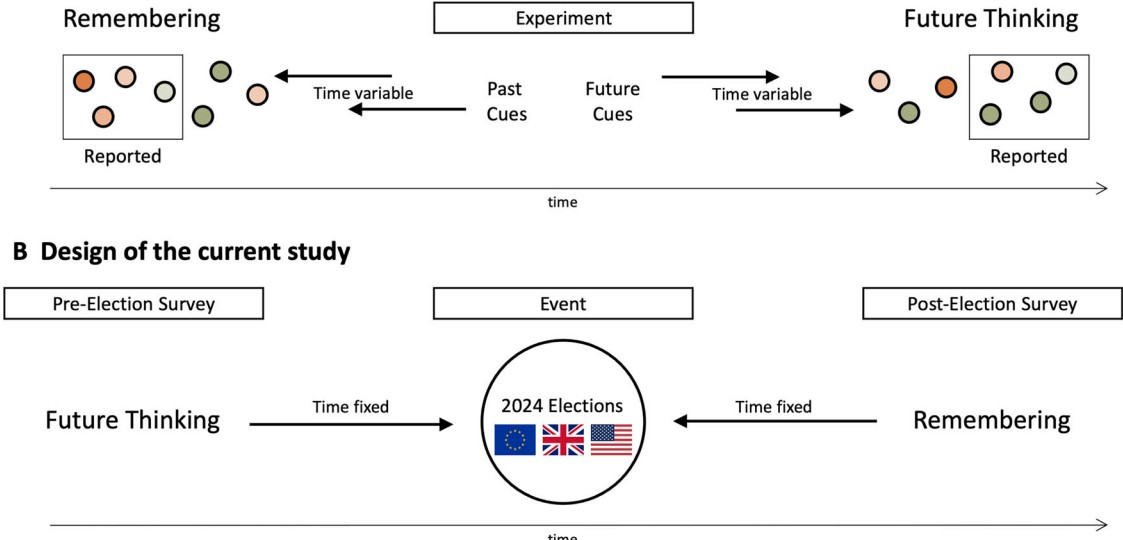

**A  Standard design of mental time travel studies**

Remembering — Experiment — Future Thinking

Past Cues ← Time variable  |  Future Cues → Time variable

Reported — time — Reported

**B  Design of the current study**

Pre-Election Survey — Event — Post-Election Survey

Future Thinking → Time fixed — 2024 Elections — Time fixed ← Remembering

time

**Fig. 1 | Standard design in mental time travel studies and the current design.**
**A** The standard approach in the field encompasses one time point, at which participants retrieve memories and imagine future events in response to cues. Consequently, the time to these events is often unconstrained and a sampling bias might arise: Participants might have other events to potentially draw on, but may select more negative ones for remembering and more positive ones for future thinking. **B** In the current study, the target event was therefore constrained to elections as an observable public event. The same participants were asked to imagine them before they happened, and to remember them afterwards. This also allowed controlling the time delays, which were matched[34] between the two processes.

Covid vaccine depends on their attitudes about these subjects[49,50]. Conversely, when opinions about an event are stable over time, memory accuracy is enhanced[33], showing that memories are influenced by current attitudes.

Collective remembering and imagining are also thought to be motivated in the sense that people select key national events that reflect positively on their country[52], and are biased towards perceiving the future of groups more central to their self as more positive[53]. They are also driven by attitudes, as in Turkey, opinions about the government systematically affected the valence of collective past and future thought[54].

The fact that these biases affect both future thinking and remembering raises the possibility that self-serving biases also influence how strongly future thinking and remembering are related.

For instance, prior research has found future events to be perceived as more important than past ones[16]. However, if an election produces a result that is more favourable to an individual than they expected, they might increase their rating of importance. This would boost current self-appraisals, as it would allow the person to feel even better about having won the election if it was ostensibly important than if it was not. Previously found similarities and differences between future thinking and remembering may therefore depend on the event itself, and the requirements it induces on a motivational self-serving memory system[55]. Self-serving biases are defined as memory distortions in line with current attitudes or feelings[11]. In the current context, these processes aimed to boost self-image would therefore emerge from a change in event appraisals, depending on the outcome of the event.

If people shift their memories compared to their future thinking to achieve self-enhancement, for instance changing their appraisal of importance, this would be to the detriment of self-consistency over time. However, an impression of self-consistency over time is one of the goals of memory biases[11]. One possible mechanism to resolve this conflict would be to distort one's memory of the way one imagined the election before it happened and to align it in accordance with current needs to boost self-esteem. For instance, election winners could erroneously believe that they always found the election highly important, even if this appraisal only emerged after having won.

Previous studies in the domain of personal remembering and future thinking have investigated these memories of the future, which are memories of the way we imagined a future event[56]. It is known that self-relevant[57], positive[58], plausible[59], and familiar[60] future simulations are remembered better, highlighting cognitive biases operating on memory and future thinking.

However, there is a lack of research testing participants on memories of the future after the event has happened, allowing the memories of imagining that future event to be checked against reality (i.e., the actual election result). It is possible that memories of the future are aligned with present attitudes, as other memories are, to increase self-consistency[11]. This is especially relevant as these memories can later be voluntarily or involuntarily reactivated[61], exerting a continued influence on ongoing cognition – reinforcing an inaccurate image of self-consistency. Therefore, this study examined whether changes between future thinking and remembering to boost self-appraisals could lead to memories of the future being shifted in the same direction, creating an illusion of consistency in attitudes over time.

In the current study, we primarily aimed to assess whether the previously identified differences in importance, valence, and vividness between future thinking and remembering would hold true when controlling for the event, asking participants to imagine the event before it happened and remember the event afterwards using a longitudinal approach. This significantly extends previous studies of personal and collective mental time travel that lacked control for the event, rather employing a cross-sectional design. Additionally, we tested whether a lack of self-congruency, resulting from self-serving changes between future thinking and remembering in response to event outcomes, would be resolved by distorting memories of the future simulations.

As events to be imagined and later remembered, we chose three nation-wide elections in 2024, namely the EU parliamentary election in Germany, the general election in the United Kingdom, and the presidential election in the United States of America. Both in the UK and USA, these elections propelled a new head of government into office and were subject to intensive media coverage in the respective country. Despite EU elections having a lower profile, the 2024 elections were seen as a 'test' for the then-ruling coalition in Germany, which increased their media visibility. All elections therefore had a high impact and their occurrence was likely to be common knowledge in each country. Importantly, elections have been previously used successfully in testing the applicability of laboratory-based effects in the real world, especially in the related field of emotion prediction and remembering[1,33,62].

Participants were asked to fill out a pre-event survey 16 to two days before the elections and were asked to imagine the election results, rating the three crucial dimensions of valence, vividness, and importance. Additionally, we obtained political identification measures as a proxy for which event outcome would be desirable. After the election, participants completed a post-event survey (keeping time periods before and after the event consistent) in which the same three measures were obtained but assessing their memory of the event (see Fig. 1B). Additionally, in the US, participants also recalled the predictions that they had given before the event, to assess memories of future thinking.

We hypothesized that, replicating previous cross-sectional findings, remembering would be associated with higher vividness, higher negativity, and lower importance than future thinking. However, we expected this to be influenced by whether participants' preferred parties or candidates won or lost the election, in that election winners would show weaker trends towards negativity and lower importance (or even a reversal of this), but a stronger trend towards higher vividness. The opposite effects were expected for election losers. Additionally, in the USA, we hypothesized that memories of the predictions would be aligned with current attitudes, reflecting self-serving biases.

## Methods
### Ethics & open science
The study protocol was accepted by the Ethics Committee of the Faculty of Psychology and Sports Science at the University of Münster (Approval Number 2024-16-MB). All participants were informed about their rights before the study and provided informed consent.

The data collections for the German EU and UK elections were conducted together with a related investigation, which was pre-registered with specific hypotheses and analysis plans for that investigation. Relevant for the current results, assessing the increased positivity in future thinking compared to remembering was a pre-registered research question as well, while no specific hypotheses were postulated for the other dimensions and for self-serving biases. However, we pre-registered all measures, target sample sizes, exclusion criteria, and precise data collection procedures, all of which applied to the results presented here, for both the EU (pre-registered on 18.05.2024; osf.io/d5knz) and UK elections (18.06.2024; osf.io/zvbh2). For the US election, the hypotheses were pre-registered based on findings from the other two studies (26.10.2024; osf.io/4c6gu), along with exclusion criteria and data collection procedures.

In addition to the pre-registrations, all questionnaires (including the German translation), as well as the data used for analysis are available on OSF (osf.io/exb3u).

### Participants
**German EU Election**. This study focused on the election of the Parliament of the European Union in Germany (09.06.2024). Participants were eligible if allowed to vote and at least 18 years old. Students at the University of Münster received course credit for each of the questionnaires, all others had the chance to enter a raffle for gift vouchers.

Based on a previous study which investigated long-term effects of surprise on memory[63], we pre-registered a target sample size of 121 responses that could be used for analysis. This sample size would allow the detection of a small effect of $f^2 = .07$. Within the community of the

**Table 1 | Overview of the samples**

|  | Germany | United Kingdom | United States |
|---|---|---|---|
| Elected body/post | EU Parliament | House of Commons | President |
| Participants at pre-election | 192 | 134 | 274 |
| Left no or faulty e-mail | 6 | 2 | 0 |
| No post-survey completed | 24 | 38 | 31 |
| Unidentifiable code | 4 | 0 | 0 |
| Failed attention check | 12 | 5 | 0 |
| Failed cheating check | 10 | 0 | - |
| **Final sample size (N)** | **136** | **89** | **243** |
| Gender | 77% Female<br>22% Male<br>1% Non-binary | 62% Female<br>38% Male | 66% Female<br>32% Male<br>1% Non-binary |
| Age | $M = 28.42$<br>$SD = 12.37$<br>Range: 18–66 | $M = 46.27$<br>$SD = 17.6$<br>Range: 18–79 | $M = 40.22$<br>$SD = 11.85$<br>Range: 19–84 |
| Students | 71% | 21% | 9% |
| Job related to politics | 2% | $M = 2.76, SD = 1.92$ | - |
| Political volunteering | 7% | $M = 2.80, SD = 1.73$ | - |
| Planned to vote | 99% | 96% | 97% |
| Turned out to vote | 97% | 97% | 96% |
| Vote shares (as assessed after election) | 46% Grüne<br>18% VOLT<br>10% CDU<br>7% SPD<br>4% Linke<br>1% FDP<br>1% BSW<br>10% Other | 52% Labour<br>25% LibDem<br>9% Green<br>4% Reform<br>2% Conservative | 52% Harris<br>43% Trump<br>1% Third-Party |

Gender was assessed using self-report. Having a job related to politics and engaging in political volunteering were not assessed in the US. In Germany, participants answered to these questions with yes or no, while UK participants used a scale from 1 = not at all to 7 = very much. The missing percentages on the vote shares are people that did not turn out to vote or spoilt their ballot.

University of Münster and through word of mouth and social media, we recruited 192 participants who finished the pre-election survey, out of which $N = 136$ provided data that could be analysed. An effect of $f^2 = 0.06$ could therefore be detected at 80% power. As pre-registered, other responses were excluded for providing no valid e-mail address for the post-election follow-up, not completing the post-election survey, not passing attention checks, or indicating that they looked up polls or election results, which they were instructed to refrain from (see Table 1 for precise numbers). The resulting sample was predominantly female (note that gender was assessed using self-report in all data collections), on average 28.42 years old ($SD = 12.37$, range: 18–66), and mainly left-leaning in their party preference (see Table 1).

**UK general election.** This study focused on the UK General Election (04.07.2024), in which members of parliament for the House of Commons are elected, and thereby effectively the Prime Minister. Anyone eligible to vote was allowed to participate in the study. Participants could enter a raffle for gift vouchers after completing each time point. Through recruitment via social media and word of mouth, 134 participants completed the pre-election questionnaire, and due to pre-registered exclusions and drop-out (see Table 1), the final sample was $N = 89$. While this fell short of the preregistered goal of 121, the achieved sample size still allowed the detection of a small effect of $f^2 = 0.09$ at 80% power. The sample was again more female, 46.27 years old on average ($SD = 17.6$, range: 18-79), and predominantly voted for the Labour Party (see Table 1).

**US presidential election.** To obtain a politically balanced sample for this study using the US presidential election (05.11.2024), participants were recruited via Prolific (www.prolific.com). As pre-registered, 274 spots were opened seven days before the election, half of which were available for the supporters of each major party. This number was higher than in the previous two data collections to account for a possible higher drop-out in a Prolific sample, and to further increase power. Participants were eligible if they were US-American, registered to vote, had previously indicated their Democratic or Republican partisanship on Prolific, were fluent in English, and had previously successfully participated in studies, following Prolific recommendations for longitudinal designs. If participants indicated no partisanship at the beginning of the survey, they were immediately screened out and the spot was opened again. Note that Prolific only allowed pre-screening for partisanship, and not for anticipated voting for Trump or Harris, so that a balanced political identification within the sample did not produce equal number of Trump and Harris voters (see Table 1). Participants were paid 0.60 GBP for completion of the pre-election survey and 0.70 GBP for the post-election survey (around 0.80 and 0.90 USD, respectively), which is above Prolific recommendations.

After exclusions, the final sample of $N = 243$ consisted of more women than other genders, with a mean age of 40.22 ($SD = 11.85$, range: 19-84). Due to some participants not voting for their party's candidate, the distribution of candidate support was slightly skewed towards Harris (see Table 1). This sample size allowed detecting a small effect of $f^2 = 0.03$ at 80% power.

**Procedure & materials**

**German EU election.** Participants could complete the pre-election survey 21 to 7 days before the EU election (20.05.2024–02.06.2024) on Pavlovia Surveys (www.pavlovia.org). They had to provide informed consent and confirm that they were at least 18 years old and eligible to participate in the EU election. To match responses across time points, participants also produced an anonymous code.

First, we asked participants whether they were planning to vote in the EU elections. For a separate investigation, we next asked them to make numeric predictions of the results. Afterwards, participants were asked which party they would vote for. We then obtained a measure of interest in politics, using three items covering general interest, interest in the EU, and news consumption. Additionally, we asked participants how much they identified with each of the nine major parties (scale from 1 to 7), which was intermixed with an attention check. These nine parties were all represented in the federal or one of the state parliaments or likely had high appeal among the target group.

The next section consisted of the three critical future thinking items. Participants were asked to imagine the outcome of the election and rate anticipated valence ("How will you feel about the outcome of the election?"; 1 = very negative – 7 = very positive), vividness ("How vividly can you picture the outcome of the election?"; 1 = not at all vividly – 7 = very vividly), and importance ("How important is the outcome of the election to you?"; 1 = not at all important – 7 = extremely important). They were asked to take the overall German EU election results into account for all items.

Lastly, participants were asked to provide demographic information (see Table 1), state if they had looked up election polls at any point during the study (which, in line with pre-registration, led to exclusion before analysis), and could leave anonymous comments. To retain anonymity, e-mail addresses for the post-election survey were collected in a separate questionnaire.

To ensure symmetry between future thinking and remembering, the time point for contacting participants for the post-election survey was determined individually. A participant who took part in the pre-survey nine days before the election was contacted eight days after the election and then had three days to respond. We contacted one day in advance to compensate for a possible bias to delay responding. However, 47% filled out the survey on the day of being contacted, resulting in a shift towards a shorter post-election interval of 0.49 days.

Participants first indicated whether they voted in the election, were asked to recall the election outcome, answered ten factual questions about the election, indicated surprise, and time spent watching election coverage – these items were administered for a separate research project and associated data are not reported in this manuscript. After indicating who they voted for, the critical three measures were repeated, now measuring how participants remembered the election outcome. Participants were asked to think back to the election outcome and asked to indicate valence ("How do you feel about the outcome of the election when you remember it now?"), vividness ("How vividly can you picture the outcome of the election when you remember it now?"), and importance ("How important was the outcome of the election to you?"), using the same scales as in the pre-election survey. After being asked to indicate whether they looked up election results during the online questionnaire (responding 'yes' led to exclusion), participants were debriefed, thanked, and forwarded to a separate questionnaire to get information on compensation.

**UK general election**. This study was an adapted replication of the EU election study. As the UK general election was called earlier than expected, German data collection was still running, so that no results-driven adaptations were made. Due to the short notice, the data collection period was also closer to the election (two to 16 days; 19.06.2024–02.07.2024). Participants tended to respond to the post-survey on the day they were contacted, leading to a forward shift of 0.52 days. Note that because the election results in the UK are announced during the night of Thursday to Friday morning, the first participants were contacted two days after Friday, so on Sunday.

After informed consent and confirming eligibility, participants indicated whether they would vote, made election outcome predictions for a separate investigation, said who they would vote for, and filled out four items on interest in politics (politics generally, UK politics, news consumption, and an added item on importance of politics). The measure of identification with each party was then conducted for the seven parties represented in the

high-profile BBC election debate on June 7. Reflecting the somewhat more binary set-up of British politics (over the past 90 years, only Labour or the Conservatives have won general elections), participants were also asked whether they wanted the Conservative or Labour party to win a majority in the House of Commons, followed up by a scale of how strong this wish was (1 = not very strongly – 3 = very strongly). These responses were then transformed to a scale of 1 = strongly Conservative – 6 = strongly Labour. Afterwards, the three crucial future thinking items were administered, using the same phrasing and scales as in the EU election study, followed by demographic information.

In the post-election questionnaire, participants indicated whether they voted, and were asked to remember how many seats each party won. After 10 factual questions, general surprise and surprise at each party's result was tested, followed by the information which party they voted for, and how much they followed election coverage. These data were collected for a separate study, and results are not reported in this manuscript. Finally, the three future thinking items, now focusing on the memory of the election result, identical to the EU election study, were repeated, followed by an honesty check and de-brief.

**US presidential election**. This study, using the US presidential election was a pre-registered follow-up on the results of the previous two studies and the data collection was exclusively for the current investigation. Instead of recruiting participants over a longer period of time, which was necessary using the convenience sampling of the previous studies, participants were recruited via Prolific and all tested exactly seven days before the election (29.10.2024). They were then invited to participate in the post-election survey exactly seven days after the election (12.11.2024), which 84% did, but responses were still accepted up to two days after this date.

After providing informed consent and confirming eligibility, participants were asked about party identification (Democrats, Republicans, or Neither), and were screened out if not aligned with one of the parties. They were then asked how much they identified with that party (1 = not very strongly – 3 = very strongly), whether they would vote, and for which candidate (Harris, Trump, Third-party candidate, spoilt ballot). After that, we asked participants who they thought would win ("If you consider it rationally now: Who do you think will win?"), who they wanted to win ("If you could make the choice: Who do you want to win?"; on both items 1 = definitely Harris, 4 = unsure, 7 = definitely Trump; Rose & Aspiras, 2020), and how fair they expected the election to be ("How fair do you think the election will be?"; 1 = definitely unfair – 7 = definitely fair). We then used the same three future thinking items on valence, vividness, and importance, phrased as in the previous studies, followed up by the items on interest in politics also used in the UK, intermixed with an attention check. Lastly, we asked for demographic information (gender, age, studying status, working status, education).

In the post-election questionnaire, participants were asked whether and for whom they voted and the items on party identification were repeated. The three items on valence, vividness, and importance, now concerning the past result, were repeated, using the same phrasing as in the German and UK study, intermixed with an attention check. Participants were also asked about surprise concerning the result, and how fair the election was.

As a new section, we also asked participants to remember their previous future simulation of the election outcomes. They were instructed to think back to the first questionnaire and to indicate which answers they had given then, not what they thought now. We asked for participants' memories of their responses to questions on who they thought would win ("Who did you think was going to win when we asked you before the election?") and who they wanted to win ("Who did you say you wanted to win when we asked you before the election?"), how fair thought the election would be ("How fair did you think the election was going to be when we asked you before the election?"), and the items on valence ("How positive or negative did you think you were going to feel about the outcome of the election when

**Fig. 2 | Pre-post changes in Valence, Vividness, and importance.** Pre-post comparisons between future thinking and remembering were carried out for each country separately (cyan = Germany, yellow = United Kingdom, red = United States). Error bars indicate the standard error ($n_{GER}$ = 136, $n_{UK}$ = 89, $n_{US}$ = 243). Significance values are indicated with asterisks: *$p$ < 0.05; **$p$ < 0.01; ***$p$ < 0.001.

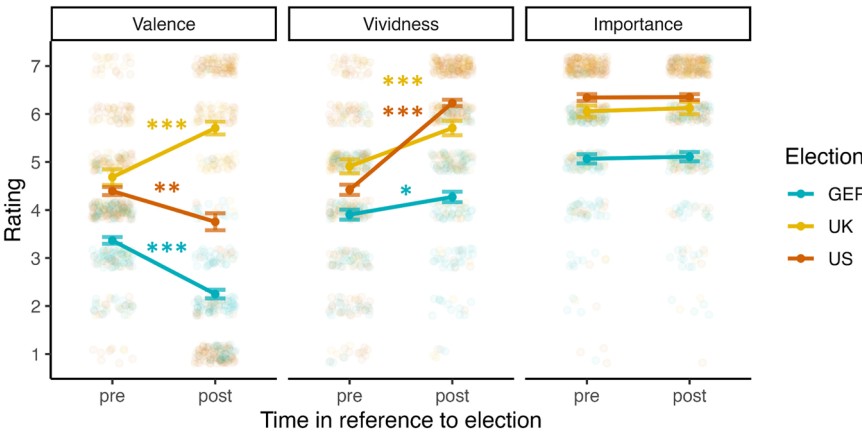

we asked you before the election?"), vividness ("How vividly did you think you could picture the outcome of the election when we asked you before the election?"), and importance ("How important did you think the outcome of the election was going to be to you when we asked you before the election?"), using the same scales as before. Lastly, participants were de-briefed and compensated via Prolific.

### Data analysis
All data was analysed using linear regression models. In addition to the variables of interest, questionnaire timing (i.e., days before and after the election that the survey was completed by a participant) as well as political interest were controlled for. All data met the necessary statistical assumptions for each analysis. In the manuscript, we report $\beta$-estimates, along with their 95% confidence intervals, and $p$-values. In the Supplementary Material, for each model, we also list $b$, $SE$, and $t$.

### Reporting summary
Further information on research design is available in the Nature Portfolio Reporting Summary linked to this article.

## Results
### Overview of the three elections
The three elections under study varied strongly in perceived consequentiality and outcome predictability. EU parliamentary elections are often rather low-profile, but turnout in the 2024 election in Germany was the second highest ever (64.7%). The two left-wing parties of the sitting federal coalition, Social Democrats (SPD) and the Greens (which was the most popular party in the sample), in line with their national polling, had disappointing results, while the third partner in the coalition, the liberal FDP, had a low but somewhat better than expected result. On the other hand, the conservative CDU and especially the far-right AfD were regarded as the election winners, the latter dominating subsequent media coverage. In the UK, the sitting Conservative Party was far behind in the polls throughout the campaign and Labour, strongly expected to win, obtained a large majority in the House of Commons (411 out of 650 seats). In the US, the race was the most unpredictable after Kamala Harris replaced Joe Biden as the Democratic candidate 107 days before the election. However, the Republican candidate Donald Trump won overwhelmingly (312 out of 538 electoral college votes), with all seven swing states in his favour and leading over Harris in the popular vote.

Almost all participants reported that they turned out to vote (Germany & UK: 97%, USA: 96%). On the measure of interest in politics, mean values were significantly above the midpoint of the scale (Germany: $M$ = 4.28, $SD$ = 1.31, $t(135)$ = 6.98, $p$ < 0.001, $d$ = 0.60; UK: $M$ = 5.86, $SD$ = 1.21, $t(88)$ = 18.45, $p$ < 0.001, $d$ = 1.96; USA: $M$ = 5.34, $SD$ = 1.27, $t(242)$ = 22.55, $p$ < 0.001, $d$ = 1.45; scale of 1–7). Interest in the events was therefore given.

The student and convenience samples in the German EU election and the UK general election were left-leaning, while we recruited equal numbers of Democrats and Republicans for the US election (see Table 1). Whether the outcome aligned with what participants would have hoped for therefore differed between the elections, with most participants being positive in the UK election, the reverse in the German EU election, and a split sample in the US.

### Comparison between future thinking and remembering
First, to test the overall similarity between future thinking and remembering, the ratings of valence, vividness, and importance were compared between the two time points, before and after each election. In this analysis, we aimed to identify general patterns without assessing self-serving biases captured by political measures. To that end, we calculated one linear model for each rating, predicted by the time point (2 levels: pre vs post). We added the measure of political interest (which was the most consistent variable in explaining expertise-related measures that are not analysed here), days to the election when completing the pre-survey (this did not apply to the US sample, where all participants completed the pre-election survey on the same day), and days since the election at post-survey as covariates. Results on these covariates are not reported in the text, but all models can be found in the Supplementary Materials (Supplementary Tables 1–9).

For all three elections, the time point had a significant influence on valence ratings, indicating differences between future thinking and remembering (see Fig. 2). Valence significantly decreased (i.e., was higher in future thinking than remembering) in Germany ($\beta$ = -0.51, 95%CI[-.61, -.41], $p$ < 0.001; pre: $M$ = 3.37, $SD$ = 0.82; post: $M$ = 2.25, $SD$ = 1.05) and the US ($\beta$ = -0.15, 95%CI[-0.24, -0.06], $p$ = 0.001, pre: $M$ = 4.40, $SD$ = 1.28, post: $M$ = 3.76, $SD$ = 2.68), but it increased for the UK ($\beta$ = 0.34, 95%CI[0.21, 0.48], $p$ < 0.001, pre: $M$ = 4.69, $SD$ = 1.55, post: $M$ = 5.71, $SD$ = 1.26). Similarly, vividness was influenced significantly by the time point, with future thinking consistently being less vivid than remembering (Germany: $\beta$ = .15, 95%CI[.03, 0.26], $p$ = 0.010, pre: $M$ = 3.90, $SD$ = 1.20, post: $M$ = 4.27, $SD$ = 1.27; UK: $\beta$ = 0.28, 95%CI[0.14, 0.41], $p$ < 0.001, pre: $M$ = 4.91, $SD$ = 1.38, post: $M$ = 5.71, $SD$ = 1.42; US: $\beta$ = 0.56, 95%CI[0.48, 0.63], $p$ < 0.001, $M$ = 4.42, $SD$ = 1.62, post: $M$ = 6.23, $SD$ = 1.03). Lastly, no significant change was found in importance before and after the event (Germany: $\beta$ = 0.02, 95% CI[-0.09, 0.13], $p$ = 0.726; UK: $\beta$ = 0.03, 95%CI[-0.10, 0.15], $p$ = 0.661; US: $\beta$ = 0.004, 95%CI[-0.08, 0.09], $p$ = 0.917; see Fig. 2).

### Correlations between changes
We next set out to investigate whether pre-post changes between the three measures were related to each other, as this would give a first indication of whether changes in importance, indicating self-serving biases, would be related to changes in vividness and valence. For all analyses, we calculated

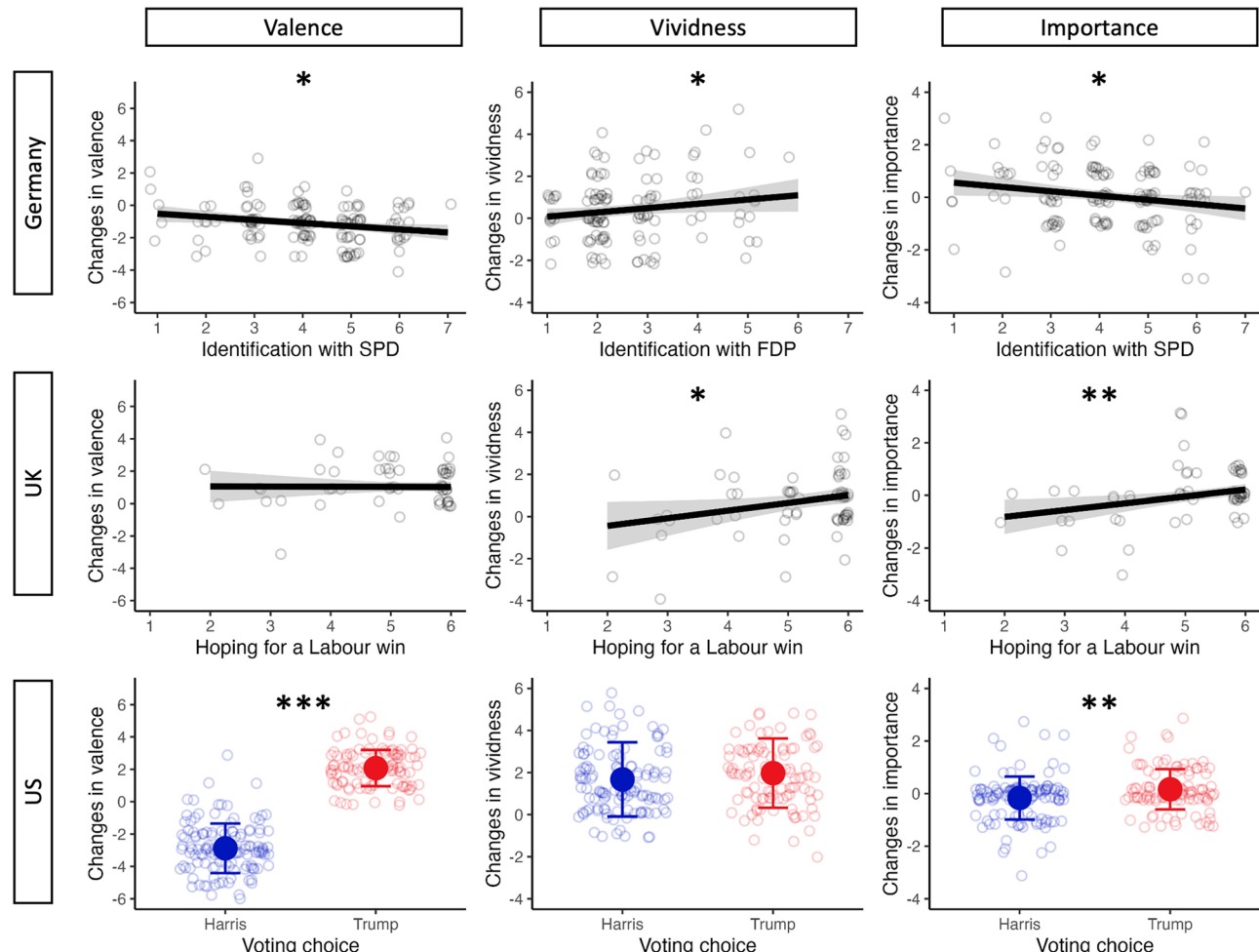

**Fig. 3 | Changes in valence, vividness, and importance depending on political affiliation.** For each election, a different measure for political affiliation was used, accounting for differences in political systems. In Germany, identifications with seven major parties were added to a model and the ones plotted emerged significant. In the UK, a scale of how much participants wanted Labour to win was used. In the US, we used the candidate that participants voted for as the political affiliation variable (blue = Harris voters, red = Trump voters). The shaded areas around the regression line and the error bars correspond to the standard error ($n_{GER}$ = 136, $n_{UK}$ = 89, $n_{US}$ = 230). Significance values are indicated with asterisks: *$p$ < 0.05; **$p$ < 0.01; ***$p$ < .001.

change scores for each participant in each measure by subtracting the pre-election response (i.e., future thinking) from the post-election value (i.e., remembering). Positive values therefore indicated increases in positivity, vividness, and importance. To control for political interest and timing of the surveys (see Supplementary Tables 10–18 for results), we specified linear models using one of the variables as the dependent variable.

Both valence and vividness changed on the sample-level in all three studies. These changes were positively correlated in the UK ($\beta$ = 0.44, 95% CI[0.24, 0.64], $p$ < 0.001) and the US ($\beta$ = 0.15, 95%CI[0.02, 0.28], $p$ = 0.022), indicating that those increasing in positivity also increased in vividness. There was no evidence for a significant correlation in Germany ($\beta$ = 0.04, 95%CI[-0.14, 0.21], $p$ = 0.677).

Even though there was no evidence for a change in importance on a sample level, individual participants did show pre-post differences on this measure. Accordingly, increases in importance were associated with increases in vividness for Germany ($\beta$ = .35, 95%CI[0.18, 0.52], $p$ < .001) and the UK ($\beta$ = 0.29, 95%CI[0.08, 0.50], $p$ = 0.007), but not in the US ($\beta$ = 0.07, 95%CI[-0.06, 0.19], $p$ = 0.293). Interestingly, the change in importance was correlated with valence in the US, indicating that when the outcome was more positive than expected, participants also increased their appraisal of how important the election was ($\beta$ = 0.20, 95%CI[0.07, 0.32], $p$ = 0.002). There was no evidence for this effect in Germany ($\beta$ = .12, 95% CI[-0.05, 0.29], $p$ = 0.158) or the UK ($\beta$ = 0.10, 95%CI[-0.12, 0.32], $p$ = 0.350).

### The effect of winning or losing on changes between future thinking and event remembering

To investigate how changes on the three variables depend on self-serving biases, we correlated them with indices of election success. In the EU elections, we asked participants about their identification with each of the nine major parties to account for the complexity in political preferences and election outcome evaluation in multi-party systems. We specified a separate linear model for each of the party identifications, while controlling for the covariates. Two parties (AfD and BSW) were excluded from this analysis as the median rating was equivalent to the low point of the scale, indicating low variance. Here, only the significant identifications are reported, but all models can be found in the Supplementary Material (Supplementary Tables 19–21).

Those that identified more strongly with the election-losing social democrats (SPD) showed stronger shifts towards more negative valence ($\beta$ = -0.23, 95%CI[-0.40, -0.06], $p$ = 0.010; see Fig. 3). Higher increases in vividness were associated with stronger identification with the liberal democrats (FDP) who did slightly better than expected ($\beta$ = 0.17, 95% CI[0.003, 0.34], $p$ = 0.046). Lastly, identification with the SPD correlated negatively with an increase in importance ($\beta$ = -0.19, 95%CI[-0.36, -0.02], $p$ = 0.026). Therefore, election winners had stronger increases in vividness, and election losers not only decreased in valence when remembering the result compared to predicting, but also ascribed lower importance to the event after the loss.

In the UK, we also asked for identification with each party, but variance in these judgements was severely limited, with only the three more left-wing parties having medians higher than one. However, the main outcome of the election is more binary, as either Labour or the Conservatives could win and form the government. We therefore asked participants in the pre-survey which of these two outcomes they would prefer and to what extent, producing a scale from 1 = definitely Conservative to 6 = definitely Labour. Even though only 8% of the sample hoped for a Conservative win, this scale also captures meaningful differences in the extent to which participants favoured a Labour win.

This measure of favoured outcome was not found to significantly predict changes in valence ($\beta = 0.01$, 95%CI[-0.21, 0.23], $p = 0.938$). However, it correlated positively with increases in vividness ($\beta = 0.25$, 95% CI[0.03, 0.47], $p = 0.026$) and importance ($\beta = 0.34$, 95%CI[0.13, 0.55], $p = 0.002$; see Fig. 3). Therefore, UK voters had a bias to increase vividness and importance of the election more strongly if the outcome was more closely aligned to their wish (see Supplementary Tables 22–24 for results on covariates).

The question of whether a participant felt like winning or losing the election is even clearer in the two-party system of the US. When asked who they hoped would win the election in the pre-survey, 86% of participants used one of the ends of the scale, indicating they definitely wanted one or the other candidate to win. This allowed us to directly compare the two groups, based on who they reported to have voted for (participants who did not vote for one of the main candidates were excluded from this analysis).

Voting behaviour strongly predicted the change in valence, with a more positive shift for Trump supporters compared to Harris voters ($\beta = 0.86$, 95%CI[0.80, 0.92], $p < 0.001$). There was no evidence for a significant difference in the change in vividness ($\beta = 0.07$, 95%CI[-0.06, 0.20], $p = 0.274$). However, the election-winning Trump supporters also showed a stronger increase in importance rating ($\beta = 0.19$, 95%CI[0.07, 0.32], $p = 0.003$; see Fig. 3).

In the binary US system, it was also more clear-cut what participants imagined when thinking about the outcome during the pre-election survey, and we assessed this directly by asking who they thought was going to win. Therefore, as an exploratory follow-up analysis, we repeated the above analysis, but while controlling for the outcome that participants imagined. For valence, the main effect of voting for Trump remained significant ($\beta = 1.46$, 95%CI[1.26, 1.67], $p < 0.001$), but there was also an effect of the outcome prediction ($\beta = 0.40$, 95%CI[0.30, 0.51], $p < 0.001$), and an interaction ($\beta = -0.91$, 95%CI[-1.17, -0.66], $p < 0.001$). Accordingly, Trump voters increased in positivity more strongly if they had predicted a Harris win, and Harris voters decreased in positivity more strongly if they had predicted a Harris win. Nevertheless, Trump voters generally still increased in positivity, even regardless of the prediction.

A similar picture emerged in the analysis of vividness, with a significant main effect of voting for Trump ($\beta = 0.92$, 95%CI[0.44, 1.40], $p < 0.001$), no significant effect of the prediction ($\beta = 0.22$, 95%CI[-0.03, 0.46], $p = 0.083$), but an interaction ($\beta = -1.04$, 95%CI[-1.63, -0.45], $p < 0.001$). Trump voters especially increased in vividness if they had predicted a Harris win, while Harris voters rather increased in vividness when they had predicted a Trump win. Lastly, for importance, the main effect observed before did not remain significant here ($\beta = 0.12$, 95%CI[-0.36, 0.60], $p = 0.630$), and neither did the outcome prediction ($\beta = -0.04$, 95%CI[-0.28, 0.21], $p = 0.764$), or the interaction ($\beta = 0.11$, 95%CI[-0.49, 0.70], $p = 0.725$). Thus, when controlling for what participants predicted, the main effects of winning or losing the elections remain significant for valence and vividness, but there was no evidence for this concerning importance (see Supplementary Tables 25–30 for full outputs of these analyses).

In summary, party identification predicts changes in the three key variables. Election winners looked back more positively than they had anticipated the election, and with higher vividness, while the opposite is true for election losers (note, however, that this pattern was not fully consistent across all three elections). Crucially, consistently across the three elections, election winners also ascribed more importance to the

event when looking back to it, while election losers reduced their appraisal of importance.

## Election fairness

Before and after the US election, we also asked participants how fair the election was going to be or was. On the whole, these ratings were higher after the election than before ($\beta = 0.19$, 95%CI[0.09, 0.28], $p < 0.001$). When analysing the change scores (remembered fairness minus imagined fairness) to assess whether self-serving biases affected participants' appraisals, it was clear that Trump supporters increased in fairness (changes: $M = 2.14$, $SD = 1.55$), while Harris supporters decreased (changes: $M = -0.63$, $SD = 2.05$), with the difference between these two changes being highly significant ($\beta = 0.60$, 95%CI[0.49, 0.70], $p < 0.001$; Supplementary Tables 31 – 32).

## Memory for future thinking

The analyses point to a role of self-enhancement biases influencing how similar future thinking and remembering are to each other. Consequently, we wanted to test how substantial changes in assessments, induced by these biases, can be reconciled with the need for self-congruency. In the US, we therefore tested whether memories of the future (i.e., memories of future simulations) would be similarly distorted by self-serving biases, and if participants would misremember their simulations to be in line with current assessments. Given the novelty of our paradigm for this field, we first tested for overall trends across the whole sample before assessing potential differences between Harris and Trump voters.

Across the sample, participants tended to overestimate how positive they had thought the election was going to be in their memories of when they imagined the event ($\beta = 0.14$, 95%CI[0.05, 0.23], $p = 0.002$; see Fig. 4), manifesting in lower values when predicting valence before the election ($M = 4.40$, $SD = 1.28$) than when asking to recall their original rating after the election ($M = 4.75$, $SD = 1.26$). They also misremembered their imagined vividness as bigger than it was ($\beta = .12$, 95%CI[0.04, 0.21], $p = 0.005$; prediction: $M = 4.42$, $SD = 1.62$; memory for prediction: $M = 4.80$, $SD = 1.42$). There was no evidence for a general misremembering of election importance judgements ($\beta = -0.01$, 95%CI[-0.09, 0.07], $p = 0.835$; prediction: $M = 6.34$, $SD = 1.08$; memory for prediction: $M = 6.33$, $SD = 0.95$).

Turning from these phenomenological items to the assessments of the election itself, participants generally misremembered how fair they thought the election would be ($\beta = 0.13$, 95%CI[0.03, 0.22], $p = 0.007$) with higher values when recalling their prediction ($M = 5.25$, $SD = 1.48$) than what they had actually predicted ($M = 4.86$, $SD = 1.61$). While there was no evidence for a misremembering of who participants *hoped* was going to win ($\beta = -0.002$, 95%CI[-0.09, 0.09], $p = 0.961$; prediction: $M = 3.67$, $SD = 2.87$; memory for prediction: $M = 3.66$, $SD = 2.87$), there was an overall shift in who they *thought* was going to win, with participants overall thinking they had been predicting a Harris win with more certainty than they actually did ($\beta = -0.12$, 95%CI[-0.21, -0.03], $p = 0.009$; prediction: $M = 4.03$, $SD = 1.64$; memory for prediction: $M = 3.60$, $SD = 1.90$; scale of 1 = definitely Harris – 7 = definitely Trump).

We next analysed differences between the voter groups in shifts between predicting and remembering the prediction. The overall bias towards misremembering higher valence was driven by Harris supporters ($\beta = -0.15$, 95%CI[-0.28, -0.02], $p = 0.021$; Harris: $M = 0.52$, $SD = 1.33$; Trump: $M = 0.16$, $SD = 1.20$) who misremembered having been more positive than they really were more strongly than Trump voters. No partisan differences in vividness ($\beta = -0.004$, 95%CI[-0.14, 0.13], $p = 0.956$; Harris: $M = 0.38$, $SD = 1.55$; Trump: $M = 0.39$, $SD = 1.57$) or importance ($\beta = 0.11$, 95%CI[-0.02, 0.24], $p = 0.089$; Harris: $M = -0.13$, $SD = 0.85$; Trump: $M = 0.16$, $SD = 1.02$) became significant.

There was no evidence for differences between Trump and Harris supporters in distorting memory for who they hoped would win ($\beta = 0.01$, 95%CI[-0.12, 0.14], $p = 0.899$; Harris: $M = -0.02$, $SD = 0.40$; Trump: $M = -0.01$, $SD = 0.78$). Interestingly, no significant difference was observed between Harris and Trump supporters in the bias of who they thought

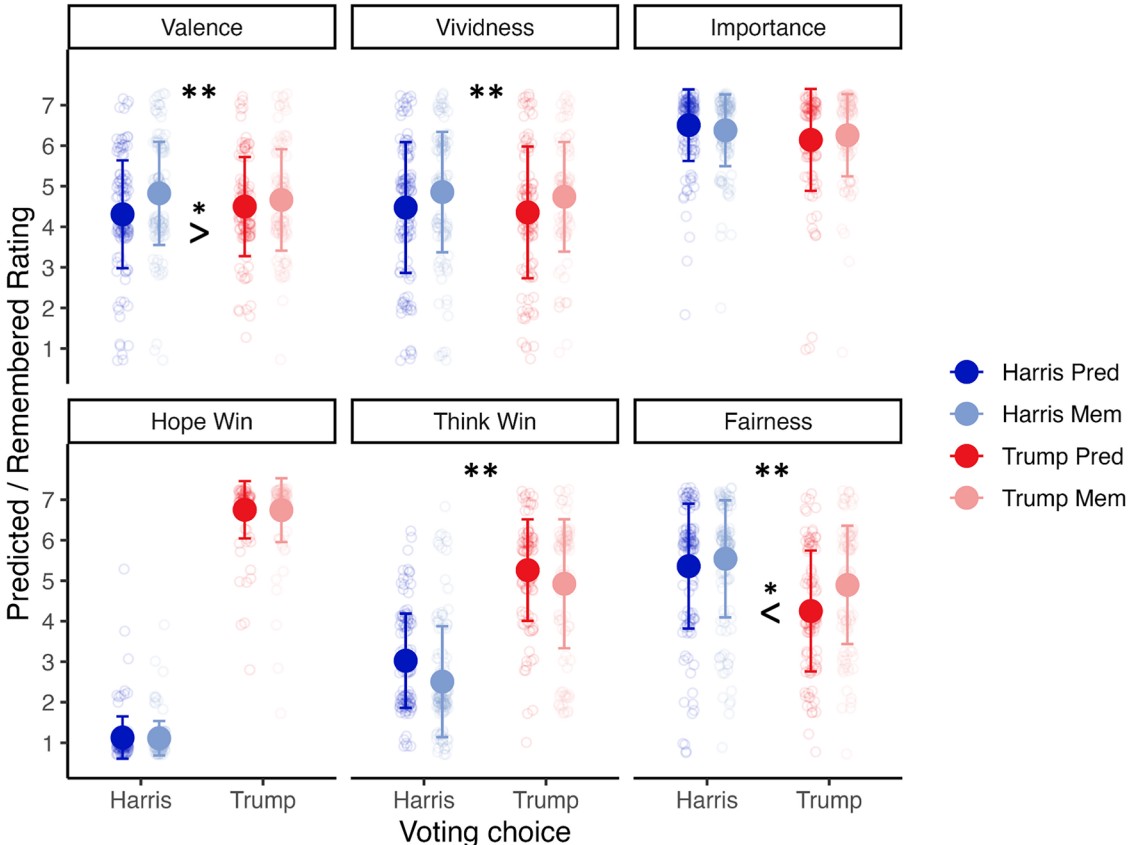

**Fig. 4 | Comparison between future thinking and memory for future thinking.** Darker colours show the predictions that participants made before the election, lighter colours the memory for the predictions (blue = Harris voters, red = Trump voters). Asterisks above the plots indicate significant changes in the overall sample, e.g., a general overestimation of positive valence. Additionally, significant differences between the partisan groups are indicated, e.g., Harris supporters more strongly overestimating how positive they were before the election. Scales reach from 1 = very negative to 7 = very positive, 1 = not at all vivid/important to 7 = very vivid/ important, 1 = definitely Harris to 7 = definitely Trump (for who they hope would win and who they think would win) and 1 = definitely unfair to 7 = definitely fair. The error bars indicate the standard error ($n = 230$). Significance values are indicated with asterisks: * $p < 0.05$; ** $p < 0.01$; *** $p < 0.001$.

would win ($\beta = 0.07$, 95%CI[-0.06, 0.21], $p = .262$; Harris: $M = -0.52$, $SD = 1.14$; Trump: $M = -0.34$, $SD = 1.24$) meaning that the sample generally tended towards thinking that they predicted a Harris win more strongly than they actually did. Turning to fairness, however, Trump supporters misremembered their fairness prediction more strongly ($\beta = 0.15$, 95% CI[0.02, 0.28], $p = 0.022$; Harris: $M = 0.18$, $SD = 1.48$; Trump: $M = 0.64$, $SD = 1.42$), thinking that they had predicted a fairer election than they really did (see Supplementary Tables 33–44 for full outputs including covariates for these analyses).

In summary, the sample generally overestimated vividness, and their certainty of who was going to win was biased towards Harris (interestingly, in the direction opposite of their current knowledge). Trump voters more strongly overestimated how fair they thought the election was going to be, and Harris voters more strongly overestimated their positivity about the outcome before the election.

## Discussion

In three longitudinal studies each surrounding a different national election, we assessed whether the outcome of an event and self-serving biases would impact differences between future thinking and remembering in valence, vividness, and importance surrounding national elections. We also tested whether they would trigger a distortion of memories of future simulations. The results of this unique approach offer three broad conclusions: First, differences between future thinking and remembering are not universal, but strongly depend on the event outcome. Second, changes between future thinking and remembering are shaped by self-serving biases, perhaps reflecting identity-driven attitudes concerning the event. Third, memories

of future simulations are also distorted by self-serving biases, so that they are in congruence with current attitudes or can explain current states. The results therefore promote understanding of the circumstances when future thinking and episodic remembering are aligned, assigning a crucial role to self-serving biases.

The first conclusion targets differences between future thinking and remembering. Previous studies have usually asked participants to generate past and future events in response to cues – at the personal[16,27,30,45] or collective level[29,31]. This captures important differences in the way people generally think about the past and future, but allows for potential confounding effects when people select different events (with different characteristics such as temporal distance or emotion) for each condition, even though other events may have been available. Here, we instead used the same specified public (and therefore observable) event for both future thinking and remembering, also allowing us to ensure temporal symmetry between conditions.

Using this alternative approach, results supported previous research showing that vividness is generally higher when remembering than when imagining future events[26,64]. This was found consistently across all three elections, even when controlling for time, and may point to the impact of the higher effort required when piecing together several memories to construct a novel future scenario[65]. In contrast, the previously found higher levels of positivity for future events[26] strongly depended on the event outcome (higher positivity when remembering in the UK, but higher positivity when imagining in Germany and the US). Lastly, while previous studies found higher importance for future events[16,27], when constraining event selection, there was no evidence for these differences on the whole-sample level (see

below for changes dependent on partisanship; note also that some studies have even found a decrease of perceived importance when remembering compared to future thinking[33,66]).

This begs the intuitive conclusion caveating the previously found differences between future thinking and remembering: Generated future events are not generally perceived as more positive and important as a product of the processes themselves but likely rather because of different selection criteria. Future thinking may be more negatively biased than remembering when the event turns out better than expected (as for the left-wing sample in the UK), but more positively biased when the event is disappointing (as for the left-wing sample in Germany). In other words, differences in valence and importance depend on *what* is being imagined and remembered. In studies using cue word methodologies, it is possible that people were more likely to optimistically imagine positive and important events. However, as shown here, they may think more realistically about the future when the choice of event is constrained to a single event.

The overall changes in the three variables between future thinking and remembering were correlated with each other. First, increases in valence were correlated with increases in vividness in the UK and USA, tallying with previous research showing that positive events are seen as more important[67–69]. This is also consistent with the finding that the future of groups that are more relevant to the self and identity are framed more positively[53]. Additionally, vividness also positively correlated with importance in Germany and the UK, and with valence in the US. It is possible that participants who increased their importance or valence ratings (which was mainly observed for election winners) also maintained higher levels of detail in the memory or were more motivated to retrieve them. As people perceive vivid memories as more accurate[33], this may be a route to partisan entrenchment of memories for an election, which can also exert an ongoing automatic influence as more vivid memories are more likely to be involuntarily retrieved[70].

These findings – that participants rated more positive than expected election results as more important than expected and also retrieve them more vividly – are therefore initial evidence that self-serving biases influence the relationship between future thinking and remembering: If election results turned out better than expected, they are also seen as more important, making the win seem more significant. To further assess this effect, we next tested directly for the relationship between political measures and changes in valence, vividness, and importance, aiming to assess the impact of one's preferred party or candidate winning or losing the election.

For valence, Trump supporters had stronger increases in positivity than Harris supporters, while for German voters, identification with the election-losing social democrats (SPD) was associated with a decrease in positivity. This once again highlights that the positivity bias in future thinking may rely on free selection of events and supports previous research which asked participants after the event to retrospectively report the expectations they had before the event[1]. Furthermore, even though vividness was robustly higher when remembering across all three samples, election winners in Germany (those more strongly identifying with the liberal FDP) and the UK (those more strongly hoping for a Labour win) showed stronger increases in vividness. As discussed above, these winning-evoked increases in vividness can give rise to subjectively higher memory accuracy judgements[33], possibly leading to partisan memory entrenchment.

Lastly, and most conclusively in terms of a self-serving bias, we found that the increase in importance attributed to the election depended on political identity. Those with low identification with the SPD, with high hopes for a Labour win, and Trump voters had stronger increases (or lower decreases) in importance between future thinking and remembering. These participants, who were more aligned with election-winning parties, therefore remembered the election as more important than they had imagined it. It is unlikely that these changes reflect objective increases in importance, as in none of the three elections would the other side winning be any less important for the future of the country than the preferred side winning. It is therefore likely that partisanship-dependent changes in importance are a product of self-serving biases. In the same vein, US-American participants adjusted their assessment of election fairness according to the outcome:

While Harris voters expected fairer elections than Trump voters, Trump voters judged fairness to be higher after the election than Harris voters.

Note that even in an exploratory follow-up analysis carried out for the US sample, in which the outcome participants imagined at the pre-election survey was controlled for, these effects were significant for both valence and vividness. Trump voters generally increased in positivity and vividness, while the degree of these changes depended on which outcome participants imagined. However, there was no evidence for partisan differences in importance changes when the imagined outcome was controlled for. Crucially, this analysis points to a contribution of the anticipation of the event, in addition to the unfolding of the event itself, on which we focused in this investigation. This further underlines that similarities and differences between future thinking and remembering are moderated by processes triggered by the nature of the event.

These findings imply that voters change their memories (and appraisals) in line with the election result, often strongly deviating from the predictions they had previously made. This fulfils a need for self-enhancement[69,71] in line with current information and opinions[55], but is to the detriment of self-consistency, as previously constructed future thoughts are overruled. Interestingly, results in the US study indicate that this can be resolved by also shifting memories of future predictions, so-called 'memories of the future'[56] in line with current attitudes.

Specifically, we found that while there was no evidence for participants misremembering their highly polarized hoped-for outcome and the generally high ascribed importance, there was an overestimation of vividness across both partisan groups, in line with the elevated vividness at the point of remembering. Similarly, memory for predicted fairness changed in line with current attitudes, as Trump voters wrongly remembered having predicted the election as fairer than they actually did – in line with their appraisal that the election was fair after knowing they had won it.

These changes are consistent with a self-congruency bias[11] where memories of previous predictions are adjusted in line with current attitudes. They are also in line with a hindsight bias, where people fail to realize to what extent their perception has been changed by an event[72]. However, there was also an opposite trend where participants adjusted their memories of the future away from their current knowledge. Instead of misremembering that they had always anticipated the outcome of a Trump win, which would reflect a hindsight bias, Harris voters showed a stronger bias towards overestimating the positivity they ascribed to the election before it happened, when asked to remember the assessment they gave before the election. Similarly, the whole sample overestimated the chances they ascribed to a Harris win over a Trump win. Despite knowing that Trump won, participants now misremembered having predicted a Harris win more strongly than they actually did.

One potential explanation testable in future research is that memories of the future are not only aligned with current attitudes but may also be influenced by current emotions[73,74], ultimately justifying and explaining them. For the Harris voters, in an attempt to explain their negative feelings just one week after the election, they perhaps assumed they had been more optimistic about the outcome (i.e., predicting higher event valence and higher chances for Harris) than they really were. This would then explain their current negative feelings better than the more ambiguous predictions that they actually gave before the election. For the Trump voters on the other hand, believing that they had been more pessimistic than they really were would conversely explain and justify their positive feelings at the moment of the post-event survey. This finding therefore shows that neither the hindsight bias[72], nor the tendency to remember positive future simulations better than negative ones[58] are ubiquitous and that both depend on the interplay between identity, current emotions, and the event.

Both congruence with current knowledge and distortion away from it, however, help to create the illusion of consistency across future thinking and remembering, and with the current situation. Encoded memories of future simulations can later be involuntarily reactivated, producing spontaneous future thought[61]. If they are distorted by self-serving biases, this bias can

therefore affect everyday cognition and feed the inaccurate perception of high self-congruency and simultaneous self-enhancement.

## Limitations

It should be noted that the key findings replicated across the three elections. This is especially important as the elections differed dramatically in their overall perceived importance and the quality and quantity of media coverage. Nevertheless, differences between the three countries may have led to not all effects being fully replicable across samples. For instance, election outcomes in Germany and the UK were highly predictable compared to the US elections, and the sample was relatively homogenous in political beliefs, limiting variance on pre-post changes. This might for instance explain why in the UK, the measure assessing which outcome participants hoped for did not predict increases in valence, possibly in combination with a dissatisfaction with the government that went beyond the ranks of Labour supporters. Conversely, the highly polarized US election was generally rated as highly important and vivid, limiting variance on these variables. As we only assessed memory of the future simulation in the US-American context, generalizability of these findings to a less polarized election remains to be established. Future studies should therefore aim to replicate these findings and extend them to other public events within and beyond politics, such as sports[63] or 'private events' such as exams.

Relatedly, this research used different sampling methods in each country. For pragmatic reasons, course credits were predominantly used in Germany, social media advertisements in the UK, and payment via Prolific in the United States. Although it is a limitation that we could not match sampling methods across studies, it is notable that we were still able to generally replicate key findings. However, these sampling approaches, yielding limited political diversity within the German and UK samples, have an impact on the generalizability of these studies individually. Importantly, however, the left-wing samples in these studies witnessed a 'successful' election in the UK and an 'unsuccessful' one in Germany, and we addressed this lack of political diversity in the balanced sample recruited in the US. Another source of diversity within the samples not controlled here was the importance of participants' political views which can influence the amount of identity fusion[36]. While the measure of political interest may have captured some of that variance, a direct measure of identity centrality[53] may further elucidate how self-serving biases moderate future thinking and remembering.

As the control analysis showed, which outcome participants imagine can interact with the unfolding of the event. However, this could only be measured in the binary US system. Future studies may therefore collect more detailed anticipated event narratives, ask participants to rank the importance of different parties' results for their overall evaluation of the election, or include additional variables capturing how the event is imagined and remembered. While in this investigation, we focused on the unfolding of the event itself, these variables would help to explore effects of the imagination.

## Conclusions

In this study, using three nation-wide elections, a longitudinal design where participants are constrained to imagining and remembering a specific public event was introduced to the field. The results have theoretical importance, showing a dynamic interplay between the event, opinions on the event, and the differences between future thinking and remembering. As the current study was situated at the intersection between collective processes, where participants remembered and imagined as a member of a political group, and personal processes, with election outcomes potentially affecting the self, the findings and the longitudinal approach can also contribute to research in collective mental time travel. Future research in the personal and collective domains should also draw on the related field of affective forecasting, where longitudinal approaches are more common. Forecasting emotions and events may be subject to different mechanisms, which may for instance yield higher vividness ratings in forecasting than remembering emotions[33], which future research should explore.

Given the setting of the study in political elections, the findings are directly applicable to the phenomenon of political polarization. This process, where groups arrive at a more extreme consensus than what was the mean of their initial opinions, can be promoted in social interactions[18,75,76]. Previous studies have also found that information associated with an ingroup is remembered better, leading to more similar accounts of events shared among ingroup members[77–81], and that selectively recalling public events can lead to collective forgetting of unmentioned events[82]. The current research demonstrates that self-serving biases congruent with one's political identity influence how we think about upcoming elections and later remember them. For example, we found that Trump supporters subsequently decided the election was fairer than they had imagined it would be. Additionally, they mistakenly believed that they had always held the view that the election would be fair. This change between future thinking and remembering, and the ensuing consistency-based misremembering of previously held beliefs can reinforce political partisanship. By remembering hoped-for events more vividly, by downplaying the importance of elections we lost, and by shifting our memories of the future in line with our current beliefs, we (possibly unintentionally and unconsciously) align our memories of events with our group identities.

Whilst biases relating to personal (as opposed to public) autobiographical events can be important at the level of the individual (e.g., reinforcing negative self-schema; [83]), understanding the way people think about public political events (such as elections) is particularly important because of their societal significance. These findings therefore also have broader implications, demonstrating how biases can shape collective memory, informed by the view that individuals have of public events. A better understanding of these mechanisms is vital, considering the power of collective memory and collective future thinking[14,84] to shape a nation's future and group behaviours from insurrection to war[85,86].

Self-serving biases may render the real facts of events elusive and ever-changing, making a cross-partisan consensus difficult. However, antidotes to polarization may exist[87,88] and future studies could explore how even a change in instructions[9] can reduce biases and contribute to less partisan future thinking and remembering. In demonstrating effects of self-serving biases on future thinking and remembering, this study highlights one psychological mechanism towards cognitive biases in the realm of political events. Future work will be necessary to replicate these findings, examine their effects on political polarisation, and explore how these biases may be reduced. Having a shared, evidence-based consensus is crucial if we are to reduce partisan violence and foster meaningful dialogue between people with contrasting political orientations.

## Data availability

The data needed to recreate all presented analyses is publicly available on OSF (osf.io/exb3u).

## Code availability

The full analysis code is publicly available on OSF (osf.io/exb3u).

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

## Acknowledgements

This work was partly supported by a grant from the Faculty of Health and Life Sciences at Oxford Brookes University, UK, and by the German Research Foundation (Deutsche Forschungsgemeinschaft) under grant number SCHU1439_10-2, project number 397530566. The funders had no role in study design, data collection and analysis, decision to publish or preparation of the manuscript. We thank Emma Davies for her helpful comments on a previous draft of this manuscript. We also thank Thea Bröring and Pia Niedrée for their help in survey preparation and Birko Boeltzig for his support in recruitment for the German sample.

## Author contributions

Marius Boeltzig: Conceptualization, Methodology, Formal analysis, Investigation, Data Curation, Writing – Original Draft, Writing – Review &

Editing, Visualization, Project administration. Ricarda I. Schubotz: Conceptualization, Methodology, Writing – Review & Editing, Funding acquisition. Scott Cole: Conceptualization, Methodology, Investigation, Writing – Review & Editing, Supervision, Funding acquisition. Clare J. Rathbone: Conceptualization, Methodology, Investigation, Writing – Review & Editing, Supervision, Funding acquisition.

## Funding

## Competing interests
The authors declare no competing interests.
