## [Transparent Peer Review file · Communications Psychology]

Self-serving biases shape the relationship between future thinking and remembering of elections

Corresponding Author: Mr Marius Boeltzig

Version 0:

Decision Letter:

Dear Mr Boeltzig,

Thank you for your patience during the peer-review process. Your manuscript titled "It was important because we won: Self-serving biases shape the relationship between future thinking and remembering of elections" has now been seen by 3 reviewers, and I include their comments at the end of this message. They find your work of interest but raised some important points. We are interested in the possibility of publishing your study in Communications Psychology, but would like to consider your responses to these concerns and assess a revised manuscript before we make a final decision on publication.

We therefore invite you to revise and resubmit your manuscript, along with a point-by-point response to the reviewers. Please highlight all changes in the manuscript text file.

Editorially, we consider it important that the revised manuscript addresses the requests for clarification and additional analysis, especially considering the outcome participants expected/their political party. Ensure your preregistration is easily accessible to reviewers and that all deviations from the preregistration are noted.

Please ensure you follow our statistical guidelines when reporting statistics (<https://www.nature.com/commpsychol/submit/submission-guidelines#statistical-guidelines>). Please note in particular our requirements for the reporting and interpretation of null-results. Non-significant findings derived from null-hypotheses significance tests should be reported in full but may not be interpreted. Where you interpret null results, this interpretation must be based on Bayes Factors or equivalence tests.

As you revise the manuscript in response to these issues, please also implement all requests in the attached Mandatory Revision Requests document. All requirements listed in this document need to be fully met, or the work will be returned to you for further revisions without peer review. This workflow is in place to increase the likelihood that the paper will be accepted for publication. It reduces the number of rounds of revision (and review) and ensures that the reviewers vet a version of the article that is compliant with journal policies. If you have any questions regarding the required revisions, please contact the journal prior to resubmission to avoid a negative outcome.

Please submit the following items:

- Revised manuscript
- Point-by-point response to the referees' comments
- Mandatory Revision Requests Table (attached).
- Cover letter (as a separate document)

- <https://www.nature.com/documents/nr-reporting-summary.pdf>>Nature Research Reporting Summary (for research Articles, Stage 2 Registered Reports, and Resources only)

via this link: Link Redacted .

** This url links to your confidential home page and associated information about manuscripts you may have submitted or are reviewing for us. If you wish to forward this email to co-authors, please delete the link to your homepage first **

Best regards,

Jennifer Bellingtier

Jennifer Bellingtier, PhD
Senior Editor
Communications Psychology

REVIEWER EXPERTISE:

episodic memory, memory biases

REVIEWER REPORTS:

Reviewer #1 (Remarks to the Author):

Thanks for the opportunity to review this interesting paper.

Unfortunately I could not access the preregistrations at the provided OSF link; the data, codebook and questionnaires were there but there was no link to the preregistration. It is possible that this may be because of recent changes to OSF structures. As a result I cannot confirm whether the data were collected and analysed as preregistered.

Please provide some justification for the selected sample size in each survey. Is the sample large enough to detect the smallest effect size of interest? Why is the sample twice as large in the US study as in the UK or German studies?

I'm a little confused about the future thinking items. When participants were asked to predict how they will feel about the outcome of the election on a scale from positive to negative, did you ask or do you know which outcome they are picturing? A voter might predict feeling very differently if they are imagining their preferred candidate winning vs. losing, so this question seems likely to be tightly linked with expectations about what the outcome is likely to be. Without knowing what participants were imagining, the valence values are meaningless, since a Labour supporter might rate their emotional response as very positive if they are imagining a Labour but very negative if they are imagining a Conservative victory.

The comparison with remembering is therefore not well-matched, since in the remembering version, participants are responding with reference to a specific outcome (e.g. the fact that Trump won) whereas for future thinking they may be imagining either a Trump or Harris victory and responding accordingly. To illustrate the point more clearly, imagine two voters who were equally happy with Trump's victory in the US election, rating their emotional response as 7 out of 7. When asked the prediction question, Voter 1 may have been imagining a Harris victory and rated their emotional response as 2 out of 7, where as Voter 2 was imagining a Trump victory and rated their response as 6 out of 7. A comparison of future thinking and memory would suggest a much more significant change in emotional state for Voter 1, but this is simply because their prediction was focused on a different outcome. As a result, there will be much more noise in the assessment of the pre-election estimates, and this seems to be borne out by the much larger standard deviations around these measures.

I think for a perfectly symmetrical comparison, participants would have to be asked something like "Imagine Trump has won the election. How will you feel?" With the current question, I'm not sure if we can rely on the data.

Reviewer #2 (Remarks to the Author):

This seems to be an interesting and important topic of study (amazing the authors obtained samples from 3 countries) but the manuscript is written in a confusing manner.

1. I can't easily find a definition of self-serving bias and how this is measured in this paper. I see in the discussion that it seems to be how important this is to the person but I am not sure how this reflects a self-serving bias.
2. Despite the fact that you use the word "recalling the event" (e.g., p. 7), participants do not seem to be recalling the election as in most (flashbulb) memory studies. They seem to be recalling how they rated a few items. Can you provide the verbatim instructions in the text? Does the use of the various samples, take away the need for a control event? I'd think it would be difficult to recall how one rated anything at two different points in time.
3. I can't figure out why this is important despite words in the paper that seem to describe this. If the study was asking about an autobiographical event that was not an election, why would it be important?
4. I can't figure out how these data support some of the claims you make, for example, regarding polarization.
5. I also cannot figure out the analyses. Why did you not regress the Time 2 rating onto the Time 1 rating and interpret the residual. Why is the method you used superior to this method?

In sum, this seems to be an impressive paper but needs to be "brought home" for the reader.

Reviewer #3 (Remarks to the Author):

The authors have undertaken an ambitious project of collecting pre-post election data in three different countries. Their work contributes to the literature of "collective future thinking" by extending research to consider the future imagination and later remembrance of a specific public event. Although I think the paper does provide new insight on collective future thinking in political context, I think it can be improved in terms of theoretical grounding and data analysis. Below you can find more details on this.

The overview authors give for the literature on episodic future thinking and self-serving biases is thorough and relevant. But the authors do not review the literature on a directly relevant concept 'collective future thinking' in detail. Since their design involves participants' imagination of election results, it perfectly fits the definition of collective future thinking provided by Szpunar and Szpunar (2016). So, I think revising the Introduction to focus less on episodic future thinking but more on collective future thinking would provide a better theoretical grounding for their paper. There are now many studies in the field of collective future thinking that focus on a wide range of phenomena that would be relevant to the current paper: valence biases (Szpunar et al., Yamashiro et al.'s, Öner et al.'s studies), perceived anomie (Ionescu et al.'s studies), perceived agency (Topcu et al.'s studies), group identification (Mert & Wang's studies, Hacibektasoglu et al.'s study) etc. The authors do not, of course, need to discuss all these studies but they can incorporate the ones that are relevant to the main arguments of this paper.

The focus of elections are quite different in all three contexts: In Germany it is EU elections, which is presumably less consequential for the country than a national election, in UK it is a parliamentary election, while in the US it is a presidential election. How do authors think these differences in election 'types' contribute to/explain the observed patterns?

Different sampling techniques are used in the three samples. A student & convenience sample in Germany (credit & raffle), a convenience sample in the UK (not paid), and a Prolific sample in the US (paid). Another difference between samples is that in the US sampling participants are pre-screened for political identification and as far as I understand they participated according to their party preferences (this should also be clarified better, did you apply quotas for Harris/Trump voters?) while in the other two samples the sampling process was not sensitive to party identification. The authors can acknowledge these differences in sampling techniques and discuss whether or not they might have contributed to the observed results.

For those who are not familiar with these three election contexts it would be helpful to summarize the actual election results at the beginning of the Results section, as that would help the readers understand the patterns better. Who won the elections in each context and who are the main opposing groups in each election context?

The first sub-section in the Results section "comparison between future thinking and remembering" and the third sub-section "The effect of winning or losing on changes between future thinking and event remembering" involve basically the same analysis at different levels: the difference of future thinking and remembering the same event. In a political context, the evaluations of an election result would inevitably be determined by participants' political leanings. So, I don't feel like the presentation of pre-post comparisons without the inclusion of political identification in the design (as the authors did in the first sub-section) would add too much to our understanding. In fact, it can be misleading because it would be highly sensitive to the sampling biases, e.g., if the sample consists mostly of winners one would inevitably see an increase in valence, while if the sample consists mostly of losers then one would inevitably see a decrease in valence. And that's exactly what happens in the observed results. So, I think the authors should consider merging these two subsections and present the comparisons for pre-post election evaluations with the inclusion of political identification in the analysis, right from the beginning.

Although I do think the results are interesting and novel, I also think that the scope is a bit limited with only three main variables of interest that are ultimately closely related to each other. The authors may consider adding variables they had measured but not included in the current analysis. People's numeric predictions of the results, for instance, could provide a better picture of how people imagine/remember the same event when they have differing predictions on the outcome of the event. I would be interested to see if the strength of the observed patterns would change based on people's predictions pre-election and how that would interact with political identification. Because as the analysis for the US sample shows one might be voting for a certain candidate while predicting the other could win.

The comment I made for the scope of the literature review in the Introduction, holds true for the Discussion as well. The authors can make more explicit links with the extant "collective future thinking" literature rather than dominantly relying on the episodic future thinking literature. As an example, in the first paragraph on pg. 22, the authors talk about research on differences between past and future events but they only refer to studies focusing on episodic memory, while there are also studies that explore the differences between collective past and future thinking.

"The finding that increases in valence and importance are associated with increases in vividness supports the idea that self-serving biases influence the relationship between future thinking and remembering." How so? Could you expand on this?

I am not sure if "self-serving biases" is the appropriate concept to discuss the changes in patterns between past-future responses. Isn't it more directly about identity-driven biases? People who identify with winners/losers adjust their ratings according to the outcome of the election based on their political identification. This can ultimately be self-serving, of course, but I feel like the connection there is more indirect and interpretive. I am not very insistent on this point, but I think the authors can consider discussing the direct/indirect links there.

In the discussion the authors can briefly discuss different sampling techniques used in three countries as a potential limitation. They do discuss the differences in political contexts across three countries, but as far as I see they did not talk about the different sampling methods used across studies in the discussion, which can be an important factor affecting cross-country comparisons.

Communications Psychology is committed to improving transparency in authorship. As part of our efforts in this direction, we are now requesting that all authors identified as 'corresponding author' create and link their Open Researcher and Contributor Identifier (ORCID) with their account on the Manuscript Tracking System prior to acceptance. ORCID helps the scientific community achieve unambiguous attribution of all scholarly contributions. You can create and link your ORCID from the home page of the Manuscript Tracking System by clicking on 'Modify my Springer Nature account' and following the instructions in the link below. Please also inform all co-authors that they can add their ORCIDs to their accounts and that they must do so prior to acceptance.
<https://www.springernature.com/gp/researchers/orcid/orcid-for-nature-research>

Version 1:

Decision Letter:

Dear Mr Boeltzig,

Your manuscript titled "Self-serving biases shape the relationship between future thinking and remembering of elections" has now been seen by our reviewers, whose comments appear below. In light of their advice I am delighted to say that we are happy, in principle, to publish a suitably revised version in Communications Psychology.

We therefore invite you to revise your paper one last time to address the remaining concerns of our reviewers and a list of editorial requests. At the same time we ask that you edit your manuscript to comply with our format requirements and to maximise the accessibility and therefore the impact of your work. When addressing the remaining requests, we ask that you retain all preregistered analyses in the main text.

EDITORIAL REQUESTS:

SUBMISSION INFORMATION:

OPEN ACCESS:

* DATA AVAILABILITY:

Link Redacted

Best regards,

Jennifer Bellingtier

Jennifer Bellingtier, PhD
Senior Editor
Communications Psychology

REVIEWERS' EXPERTISE:

episodic memory, memory biases

REVIEWERS' COMMENTS:

Reviewer #1 (Remarks to the Author):

The authors have addressed all my concerns satisfactorily, and I am happy to recommend publication.

Reviewer #2 (Remarks to the Author):

The revised version of the manuscript is much clearer. This manuscript makes a substantial contribution to the literature. I have only two minor comments.

1. If you are aware of the estimated payment in US dollars at the time of data collection, please add this information.
2. On page 30 under limitations, please consider changing "highly polarized US elections were..." to "highly polarized US election was..."

Reviewer #3 (Remarks to the Author):

The authors have done a good and thorough work in addressing the previously raised points. I think the article now is ready for publication. I only have 2 minor points, which can easily be addressed in a small revision:

1. The section on memory for Future Thinking can be revised to be more straightforward. Specifically, the authors can directly report the analysis with voter choice taken into consideration, as Figure 4 reflects. In the current version, they do the analysis on the comparison of future ratings and memory-for-future ratings without voter choice as a factor, which is then followed by the same analysis with voter choices included as a factor. This presentation unnecessarily lengthens the results and makes it difficult to follow. Especially in a binary political context like the US, the analysis should include voter choice right from the beginning.
2. One result that seemed intriguing and counterintuitive is that in the UK the measure of favored outcome did not predict changes in valence. I closely read the discussion and could not find how authors interpret this result. What could account for this result?

Responses to Reviewers' Comments

Reviewer #1 (Remarks to the Author)

Thanks for the opportunity to review this interesting paper. Unfortunately I could not access the preregistrations at the provided OSF link; the data, codebook and questionnaires were there but there was no link to the preregistration. It is possible that this may be because of recent changes to OSF structures. As a result I cannot confirm whether the data were collected and analysed as preregistered.

We apologize for this issue. The pre-registrations used to be automatically connected to the project, but this seems to have changed with the recent update. The full project with all data can be accessed via osf.io/exb3u and the pre-registrations are osf.io/d5knz (GER), osf.io/zvvh2 (UK), and osf.io/4c6gu (US). These direct links are also added to the manuscript (pp. 9-10).

Please provide some justification for the selected sample size in each survey. Is the sample large enough to detect the smallest effect size of interest? Why is the sample twice as large in the US study as in the UK or German studies?

The initial determination of desired sample size for the studies in Germany and the UK ($N = 121$) was based on a study by Antony et al. (2023), which focused on memory effects of long-term prediction errors. This was the focus of the study that shared the data collection with the current one. As we could not build on prior studies in our investigation, with the novel approach of directly comparing future thinking and remembering within the same event, we used this paper to inform the desired sample size.

This target sample size of $N = 121$ allowed the detection of even a small effect of $f^2 = .07$ with 80% power. As the obtained sample size exceeded the goal in Germany, the smallest detectable effect size at 80% power was $f^2 = .06$, while it was $f^2 = .09$ in the UK, where we did recruit 134 participants for the pre-election survey, but drop-out was higher than in the German study. Nevertheless, in both samples, even small effects could be detected at high power.

The US sample is twice as big because we obtained funding for this follow-up study based on the results from the EU and UK elections. This allowed us to run a pre-registered replication with even higher power and with a politically balanced sample, also accounting for a presumably higher drop-out in the Prolific setting. With the obtained sample size, the smallest effect size detectable at 80% power was $f^2 = .03$.

We have added information on power to the respective Participants sections of each study (pp. 10-12). We have also added considerations on the diversity and size of the samples to the Limitations section (pp. 30-31).

I'm a little confused about the future thinking items. When participants were asked to predict how they will feel about the outcome of the election on a scale from positive to negative, did you ask or do you know which outcome they are picturing? A voter might predict feeling very differently if they are imagining their preferred candidate winning vs. losing, so this question seems likely to be tightly linked with expectations about what the outcome is likely to be. Without knowing what participants were imagining, the valence values are meaningless, since a Labour supporter might rate their emotional response as very positive if they are imagining a Labour but very negative if they are imagining a Conservative victory.

The comparison with remembering is therefore not well-matched, since in the remembering version, participants are responding with reference to a specific outcome (e.g. the fact that Trump won) whereas for future thinking they may be imagining either a Trump or Harris victory and responding accordingly. To illustrate the point more clearly, imagine two voters who were equally happy with Trump's victory in the US election, rating their emotional response as 7 out of 7. When asked the prediction question, Voter 1 may have been imagining a Harris victory and rated their emotional response as 2 out of 7, whereas Voter 2 was imagining a Trump victory and rated their response as 6 out of 7. A comparison of future thinking and memory would suggest a much more significant change in emotional state for Voter 1, but this is simply because their prediction was focused on a different outcome. As a result, there will be much more noise in the assessment of the pre-election estimates, and this seems to be borne out by the much larger standard deviations around these measures.

I think for a perfectly symmetrical comparison, participants would have to be asked something like "Imagine Trump has won the election. How will you feel?" With the current question, I'm not sure if we can rely on the data.

We thank the Reviewer for this thoughtful point. Future work could indeed constrain which outcome participants are asked to imagine, as this could especially interact with the memories for the simulation. However, in this study, with its unique longitudinal design focused on a single event, we did not want to constrain participants' future thinking, but were instead interested in how they actually felt towards a naturally occurring and highly consequential event, and how the unfolding of the event would impact differences between future thinking and remembering.

The lack of matching between future thinking and remembering, where only the latter process is carried out with knowledge of the actual outcome is therefore the naturally occurring situation. Crucially, the process that the Reviewer describes is a central conclusion of the investigation: Instead of stable differences between future thinking and remembering, which

previous studies using the event generation method have implied, we find that the differences between the two events depend on the event itself – and consequently also what participants imagine at the pre-election survey. Imagining a worse outcome than reality, and then increasing valence judgements is therefore part of the processes we aimed to capture.

However, it is true that more noise is inherent in the pre-election survey. Therefore, to further pursue the interesting point raised by the Reviewer, also in response to a similar comment by Reviewer 3, we added an exploratory post-hoc analysis (pp. 21-22) where we control for which outcome participants imagine in the US election. In that election, in contrast to the other two, the prediction is clear-cut (i.e., only one of two possible outcomes), and we assessed it by asking participants who they thought was going to win before the election. Crucially, concerning valence, which the Reviewer focused on, even when adding this variable to the model, the main effect of partisanship remained significant. This indicates that Trump voters generally increased in their appraisal of positive valence, and Harris voters generally decreased. Only in addition to this significant main effect, there were main effects of whether participants voted for Trump or Harris, and an interaction. Specifically, those Trump voters that had imagined a Harris win increased more strongly in appraisal of positivity, and those Harris voters that had imagined a Harris win decreased it more strongly.

Consequently, the Reviewer is right that what participants predict has an influence on the change in valence between future thinking and remembering, but in addition to this effect, the variable of partisanship, which we had focused on initially, remains significant. We discuss this analysis accordingly and believe that it further strengthens the main outcome of our investigation: differences between future thinking and remembering are contingent on the event itself. While we had focused on the unfolding of the event, the Reviewer's comment focuses on the complementary influence of outcome anticipation, which we believe is a fruitful area for further research as well. We therefore also raise this point in the Limitations section (p. 31).

Reviewer #2 (Remarks to the Author):

This seems to be an interesting and important topic of study (amazing the authors obtained samples from 3 countries) but the manuscript is written in a confusing manner.

We thank the Reviewer for this comment and have revised the manuscript to improve clarity and readability, in line with their comments.

1. I can't easily find a definition of self-serving bias and how this is measured in this paper. I see in the discussion that it seems to be how important this is to the person but I am not sure how this reflects a self-serving bias.

We thank the Reviewer for pointing that out. The self-serving bias in our design emerges from changes between pre-election survey and post-election survey, based on the unfolding of the event itself. For instance, increases in importance when one's party wins the election, but decreases in importance when it loses, would constitute a self-serving bias. We now make this definition clearer in the Introduction: "Self-serving biases are defined as memory distortions in line with current attitudes or feelings (Schacter et al., 2023). In the current context, these processes aimed to boost self-image would therefore emerge from a change in event appraisals, depending on the outcome of the event." (p.7)

2. Despite the fact that you use the word "recalling the event" (e.g., p. 7), participants do not seem to be recalling the election as in most (flashbulb) memory studies. They seem to be recalling how they rated a few items. Can you provide the verbatim instructions in the text? Does the use of the various samples, take away the need for a control event? I'd think it would be difficult to recall how one rated anything at two different points in time.

We have added the verbatim instructions to the Methods section (pp. 13-16), as they were used directly in the UK and US experiments. The German translations can be accessed via OSF. This suggestion will help the reader to quickly grasp what exactly was required from the participants.

The Reviewer is right that participants were asked at the post-election survey to recall how they had predicted the event at the pre-election survey – but only when it comes to memory for the future simulation, which was only assessed in the US sample. These results can be found on page 23 in a section called "Memory for Future Thinking" and in Figure 4. Here, participants were asked to recall what their prediction had been before the election, revealing that their memory of how they had predicted the election was also distorted in a self-serving way. Even though this amounted to asking participants to remember how they had rated these items before, their responses in the pre-election survey that they were asked to recall, were reflective of their predictions before the election. What they were asked to remember was therefore not merely their response to an item in an online survey, but their anticipation of a major political event, likely to shape their own and their country's future.

However, the other analyses indeed focused on a comparison between future thinking and remembering the event, so that the label of "recalling the event" is correct. Participants were asked to predict valence, vividness, and importance of the outcome in the pre-election survey (future thinking), and to think back to the result in the post-election survey and rate valence, vividness, and importance of the event when remembering it. These items are therefore not a recall of previous ratings, but an assessment and remembering of the event from today's standpoint. We hope that the added verbatim instructions, together with additional clarification

in the Methods section (p. 13), and a new Figure 1 depicting the paradigm, help to make the design clearer.

Against this background, we do not believe that a control event is needed to interpret the results. As we sampled participants with differing political attitudes and partisanship, we could directly measure the effect of political identity on differences between future thinking and remembering, practically offering a control in itself. However, we do agree that future studies should increase generalizability of the study by using other events than political elections, a point that is now also made in the Limitations section.

3. I can't figure out why this is important despite words in the paper that seem to describe this. If the study was asking about an autobiographical event that was not an election, why would it be important?

First, the results are theoretically relevant, showing that differences between future thinking and remembering are not universal, but depend on the event in question, a point that is elaborated on in the Discussion (pp. 26-27).

Additionally, there is a high practical relevance. Biases relating to personal (as opposed to public) autobiographical events can be important at the level of the individual – they may, for example, reinforce negative self-schema (e.g. Beck, 1967), or conversely increase self-appraisals by evaluating events depending on how positively they unfolded for the individual. Understanding the way people think about important, public political events (such as elections) is particularly important as these biases may help to explain the development of polarized views within society and relate to political assessments more broadly. We have added to the discussion (p. 32) to make this clearer.

4. I can't figure out how these data support some of the claims you make, for example, regarding polarization.

We agree that this could have been made clearer. We have now added to the Discussion (see p. 32) to explain how these biases might lead to polarization.

5. I also cannot figure out the analyses. Why did you not regress the Time 2 rating onto the Time 1 rating and interpret the residual. Why is the method you used superior to this method?

We thank the Reviewer for this suggestion. Indeed, the analysis they suggest is an alternative to the approach we used. This suggestion would put an emphasis on the Time 2 values, accounting for (i.e., regressing out) baseline levels at Time 1. However, in the paper, we are

interested in precisely the pre-post comparison between future thinking and remembering. We therefore opted to use change ratings, which were computed by subtracting the pre-election ratings from the post-election ratings. These ratings directly reflect the *change* in the evaluation between future thinking and remembering and can be easily interpreted as such in the descriptive values and figures. Instead of placing emphasis on the Time 2 values, while accounting for Time 1 values, we therefore directly and visibly model the change between time points, and therefore directions of mental time travel. Statistically, this approach does not produce lower power than the approach suggested by the Reviewer, but has the advantage of directly aligning with the goal of the paper.

In sum, this seems to be an impressive paper but needs to be "brought home" for the reader.

We thank the Reviewer for this assessment and hope that the changes to the manuscript, in line with their suggestions, improved the clarity of the paper.

Reviewer #3 (Remarks to the Author):

The authors have undertaken an ambitious project of collecting pre-post election data in three different countries. Their work contributes to the literature of "collective future thinking" by extending research to consider the future imagination and later remembrance of a specific public event. Although I think the paper does provide new insight on collective future thinking in political context, I think it can be improved in terms of theoretical grounding and data analysis. Below you can find more details on this.

The overview authors give for the literature on episodic future thinking and self-serving biases is thorough and relevant. But the authors do not review the literature on a directly relevant concept 'collective future thinking' in detail. Since their design involves participants' imagination of election results, it perfectly fits the definition of collective future thinking provided by Szpunar and Szpunar (2016). So, I think revising the Introduction to focus less on episodic future thinking but more on collective future thinking would provide a better theoretical grounding for their paper. There are now many studies in the field of collective future thinking that focus on a wide range of phenomena that would be relevant to the current paper: valence biases (Szpunar et al., Yamashiro et al.'s, Öner et al.'s studies), perceived anomie (Ionescu et al.'s studies), perceived agency (Topcu et al.'s studies), group identification (Mert & Wang's studies, Hacibektasoglu et al.'s study) etc. The authors do not, of course, need to discuss all these

studies but they can incorporate the ones that are relevant to the main arguments of this paper.

We appreciate the Reviewer's point and agree that there is relevance to the growing literature on collective mental time travel. We have adapted the Introduction to embed the concept of collective mental time travel as holding key relevance in the study of elections (we also cite key studies that are relevant to remembering political events, for example Cheriet's work). As well as theoretical links, we added key studies that add to the findings from personal mental time travel – for example, on the correlation between past and future thinking, key differences, and ways they are cued. We hope this extension to the collective literature clarifies these links, so that the reader is able to see the relevance to collective as well as personal future thinking.

To give two examples of these changes:

- p. 4. "At the collective level, reviews have concluded that collective future thinking is more negative, and less specific, than collective remembering (Liu & Szpunar, 2023; Topcu & Hirst, 2022)."
- p. 7: "Collective remembering and imagining are also thought to be motivated in the sense that people select key national events that reflect positively on their country (Yamashiro & Roediger, 2019), and are biased towards perceiving the future of groups more central to their self as more positive (Berntsen & Rubin, 2024). They are also driven by attitudes, as in Turkey, opinions about the government systematically affected the valence of collective past and future thought (Hacıbektaşoğlu et al., 2023)."

The focus of elections are quite different in all three contexts: In Germany it is EU elections, which is presumably less consequential for the country than a national election, in UK it is a parliamentary election, while in the US it is a presidential election. How do authors think these differences in election 'types' contribute to/explain the observed patterns?

We agree with the assessment that the three elections differ in perceived consequentiality and believe that this is a strength of our approach. As we ran the analyses comparing future thinking and remembering, as well as their dependence on political identity measures, in all three countries, we increase generalizability and robustness compared to a situation where only one election is used. It is often the case in comparable studies that one election (often in the US) is studied, leaving it open how the results would generalize to a less polarized and binary political set-up. The key results replicate across the three different political systems, and we interpret the differences in outcomes against the backdrop of different situations. For instance, the fact that positive valence increases in the UK and the US, but decreases in Germany, can be explained by the samples reacting to the specific election outcomes.

The only exception to this approach is the analysis of memory for future simulations, as

we only assessed this in the US study. It can therefore not be ruled out that the memory distortions seen there would be attenuated in a less consequential or polarized political election. We have added this point to the Limitations section (p.30), and also draw the reader's attention to the differences between the elections at the beginning of the Results section (pp. 16 – 17), where we now (in line with the Reviewer's later comment) provide context on the elections.

Different sampling techniques are used in the three samples. A student & convenience sample in Germany (credit & raffle), a convenience sample in the UK (not paid), and a Prolific sample in the US (paid). Another difference between samples is that in the US sampling participants are pre-screened for political identification and as far as I understand they participated according to their party preferences (this should also be clarified better, did you apply quotas for Harris/Trump voters?) while in the other two samples the sampling process was not sensitive to party identification. The authors can acknowledge these differences in sampling techniques and discuss whether or not they might have contributed to the observed results.

We have added clarification on the sampling process on Prolific. The Reviewer is right that we applied quotas only for party identification, and not for voting intention. Prolific has a range of data on their participants which they can provide voluntarily, but who they planned to vote for in the 2024 election was not registered there. We therefore could only apply quotas for partisanship, which were confirmed in the study itself, and rely on rough alignment between partisanship and voting intention. As displayed in Table 1, the final sample comprised 52% Harris voters, and 43% Trump voters, as there was a share of Republicans who voted for Harris (or not at all). We have now made this procedure more transparent in the Participants section (p. 11).

Additionally, we have acknowledged differences in sampling in the Limitations section (pp. 30-31). While we interpret differences between the three studies carefully against their electoral backdrop, more similar samples and recruitment strategies are a valuable suggestion for future research.

For those who are not familiar with these three election contexts it would be helpful to summarize the actual election results at the beginning of the Results section, as that would help the readers understand the patterns better. Who won the elections in each context and who are the main opposing groups in each election context?

We thank the Reviewer for this suggestion. We moved the background information on the three elections that was presented in the Supplementary Materials to the beginning of the Results section (p. 11) so that readers can more easily interpret the found effects. All parties that are relevant during the analysis are mentioned in that section.

The first sub-section in the Results section “comparison between future thinking and remembering” and the third sub-section “The effect of winning or losing on changes between future thinking and event remembering” involve basically the same analysis at different levels: the difference of future thinking and remembering the same event. In a political context, the evaluations of an election result would inevitably be determined by participants’ political leanings. So, I don’t feel like the presentation of pre-post comparisons without the inclusion of political identification in the design (as the authors did in the first sub-section) would add too much to our understanding. In fact, it can be misleading because it would be highly sensitive to the sampling biases, e.g., if the sample consists mostly of winners one would inevitably see an increase in valence, while if the sample consists mostly of losers then one would inevitably see a decrease in valence. And that’s exactly what happens in the observed results. So, I think the authors should consider merging these two subsections and present the comparisons for pre-post election evaluations with the inclusion of political identification in the analysis, right from the beginning.

We thank the Reviewer for this point. They are right that the direction and magnitude of pre-post comparisons will be strongly influenced by political leaning and that this is what subsequent analyses focus on.

We have included this analysis as a simple and straight-forward comparison between future thinking and remembering. The motivation for this was to directly evaluate the claims of lower valence, lower importance, and higher vividness when remembering compared to thinking about the future that were raised in previous studies (Berntsen & Bohn, 2010; Morton & MacLeod, 2023; Rasmussen & Berntsen, 2013). In our study, we controlled the event to assess whether these differences still hold. The analysis in question therefore helps to make the point that the differences between processes depend on the events themselves – as is seen for instance in the valence ratings changing in opposite directions across the three sets of data collected. Crucially, these analyses also show that, across the sample, vividness is indeed higher when remembering, and that there are no overall changes in importance. These points would be lost if we immediately added political measures, as we would then only be able to conclude whether there are partisan differences in the changes, but not whether there is a change overall. An omission of this analysis would therefore cloud the intuitive, but theoretically important conclusion that differences between remembering and future thinking depend on the event itself.

We have added clarity in the Results section as to what this analysis encapsulates (p.17). For the reasons outlined above, we would prefer to keep the analysis in the manuscript. However, if the Reviewer and Editor agree that removing the analysis would be beneficial, we are happy to comply.

Although I do think the results are interesting and novel, I also think that the scope is a bit limited with only three main variables of interest that are ultimately closely related to each other. The authors may consider adding variables they had measured but not included in the current analysis. People's numeric predictions of the results, for instance, could provide a better picture of how people imagine/remember the same event when they have differing predictions on the outcome of the event. I would be interested to see if the strength of the observed patterns would change based on people's predictions pre-election and how that would interact with political identification. Because as the analysis for the US sample shows one might be voting for a certain candidate while predicting the other could win.

We thank the Reviewer for the interesting idea to control for what people actually imagined when making a prediction (see also the response to Reviewer 1's similar point). For the German EU election, this analysis is unfortunately not straight-forward. How participants imagined the election there is likely to depend on a blend of predictions concerning possibly up to nine parties, and how much weight each one gets in this blend will differ from participant to participant, in a complex interplay between favouritism for one's preferred party and antipathy against other parties. This blend at the pre-election survey will then interact with the equally complex outcome. Therefore, as there is no obvious way of collapsing across the nine percentage predictions and results without making strong assumptions (which were not pre-registered), we refrained from conducting such an analysis for the German sample. The same is true for the UK, where even more parties gained seats in the House of Commons. Additionally, for the most important outcome, namely which party will gain a majority, only two participants predicted a Conservative majority in this highly predictable election which Labour was poised to, and did indeed, win overwhelmingly.

However, in the US, it is much more clear-cut what participants imagine and we did assess this explicitly with the item asking about who participants think will win the election. We therefore added a follow-up analysis to control for this outcome prediction (pp. 21-22). Interestingly, the partisan differences in changes of valence and vividness remain significant, even if taking into account whether participants predict a Harris or Trump win before the election. However, the effect of importance loses significance. We now discuss these results in the manuscript (p. 28) and believe that it further strengthens the main claim of the paper, namely that differences between future thinking and remembering are moderated by the event itself. While we focused on the unfolding of the event, via the political identity measures, this control analysis shines an additional light on the imagined outcome of the event. Consequently, future studies should explore this interplay between participants' predictions and the unfolding of the event further.

In addition to reporting and discussing this analysis, we expanded on the point that had already been raised in the Discussion, focusing on the possibility that collecting imagination protocols may be an interesting possibility for future research.

The comment I made for the scope of the literature review in the Introduction, holds true for the Discussion as well. The authors can make more explicit links with the extant “collective future thinking” literature rather than dominantly relying on the episodic future thinking literature. As an example, in the first paragraph on pg. 22, the authors talk about research on differences between past and future events but they only refer to studies focusing on episodic memory, while there are also studies that explore the differences between collective past and future thinking.

We appreciate the Reviewer raising the relevance of collective mental time travel. We have now extended the Discussion by including several theoretical and empirical citations from the CMTT literature. Below are examples of where we have added parts in this section. We hope these additions make relevant links to this emerging literature.

Here are two examples from the changes made:

- p. 26: “Previous studies have usually asked participants to generate past and future events in response to cues – at the personal (Berntsen & Bohn, 2010; Cole & Berntsen, 2015; Özbek et al., 2020; Rasmussen & Berntsen, 2013) or collective level (Liu & Szpunar, 2023; Mert et al., 2023).”
- p. 27: “This is also consistent with findings by Berntsen and Rubin (2024) that the future of groups that are more relevant to the self and identity are framed more positively.”

The finding that increases in valence and importance are associated with increases in vividness supports the idea that self-serving biases influence the relationship between future thinking and remembering.” How so? Could you expand on this?

We have expanded this section in the text to make our reasoning clearer (p.27). The sentence was an interim summary of the correlations discussed in the preceding paragraph, which is now spelled out more clearly. Instead of a universal drop in importance and valence, and increase in vividness, the correlations between these effects indicate that they are related to each other, and as the judgement of importance depends on the event itself, moderated by self-serving biases.

I am not sure if “self-serving biases” is the appropriate concept to discuss the changes in patterns between past-future responses. Isn’t it more directly about identity-driven biases? People who identify with winners/losers adjust their ratings according to the

outcome of the election based on their political identification. This can ultimately be self-serving, of course, but I feel like the connection there is more indirect and interpretive. I am not very insistent on this point, but I think the authors can consider discussing the direct/indirect links there.

We thank the Reviewer for pointing this out and understand the reasoning that the changes between future thinking and remembering being moderated by political identification variables might not directly be self-serving, but rather identity-driven or group-serving. The broadening of scope to collective processes, suggested in an earlier comment by this Reviewer, will help to address this issue and make the intersection between individual and collective processes clearer.

Crucially, we have understood the effects as self-serving on the basis of identity fusion theory (Swann et al., 2012). This theory postulates that group memberships can merge into the self, producing a strong feeling of being one with the group. Prior research has found that this process is especially strong in the lead-up to elections, in that case the 2016 US presidential election (Misch et al., 2018). We therefore believe that partisan interests can become personal interests in the climate of elections. However, we recognize that the amount of identity fusion might differ between participants. However, while we did not measure this in the current study, the inclusion of our multi-item measure of political interest may have accounted for substantial parts of this variance.

To address this issue, we have added explicit clarification on the connection between self and group when identity fusion theory is first introduced (p. 6). Additionally, in the context of discussing sampling strategies, we also included considerations on differing amounts of identity fusion (p. 31).

In the discussion the authors can briefly discuss different sampling techniques used in three countries as a potential limitation. They do discuss the differences in political contexts across three countries, but as far as I see they did not talk about the different sampling methods used across studies in the discussion, which can be an important factor affecting cross-country comparisons.

We agree that this is an important point and have now included this explicitly in the Limitations section (pp. 30-31). Crucially, we interpret the replicable findings across the three countries together and interpret differences against the electoral backdrop, mitigating the problem that the samples differed from each other.

Responses to Reviewers' Comments

Reviewer #1 (Remarks to the Author)

The authors have addressed all my concerns satisfactorily, and I am happy to recommend publication.

We thank the Reviewer for their favourable evaluation and their valuable comments on the first version of this manuscript.

Reviewer #2 (Remarks to the Author)

The revised version of the manuscript is much clearer. This manuscript makes a substantial contribution to the literature. I have only two minor comments.

We are glad that the Reviewer finds the manuscript clearer now and we thank them for their comments that shaped this new version.

1. If you are aware of the estimated payment in US dollars at the time of data collection, please add this information.

We have added the estimated payment in US dollars to the Participants section (p. 10-11).

2. On page 30 under limitations, please consider changing "highly polarized US elections were..." to "highly polarized US election was..."

We have changed this to the singular as suggested (p. 28).

Reviewer #3 (Remarks to the Author):

The authors have done a good and thorough work in addressing the previously raised points. I think the article now is ready for publication. I only have 2 minor points, which can easily be addressed in a small revision:

We thank the Reviewer for this evaluation and their thorough comments on the first version of the manuscript.

1. The section on memory for Future Thinking can be revised to be more straightforward. Specifically, the authors can directly report the analysis with voter choice taken into

consideration, as Figure 4 reflects. In the current version, they do the analysis on the comparison of future ratings and memory-for-future ratings without voter choice as a factor, which is then followed by the same analysis with voter choices included as a factor. This presentation unnecessarily lengthens the results and makes it difficult to follow. Especially in a binary political context like the US, the analysis should include voter choice right from the beginning.

We understand this concern for a shorter results section, especially given the polarized backdrop of the election. However, we think that the analysis that does not take voting choice into account is a crucial part of the manuscript. This is because our paradigm was novel to the field and therefore allowed testing for general trends in memory for simulations before adding the political groups. We have added this clarification to the manuscript: “Given the novelty of our paradigm for this field, we first tested for overall trends across the whole sample before assessing potential differences between Harris and Trump voters.” (p. 21)

Interestingly, despite potentially strong effects of partisanship, there are several trends that actually are shared across partisan divides: Both voter groups overestimate their vividness and, remarkably, both groups think they had predicted the outcome closer to a Harris win than they actually did. We believe that these effects should be reported and believe that an overall future thinking vs memory of future thinking comparison is the most straight-forward way to do so.

2. One result that seemed intriguing and counterintuitive is that in the UK the measure of favored outcome did not predict changes in valence. I closely read the discussion and could not find how authors interpret this result. What could account for this result?

We thank the Reviewer for pointing this out. Indeed, this result is counter-intuitive and not in line with the changes in valence observed in both Germany and the US. However, we believe that this is a product of our sample and the dynamics of the election. Our sample was predominantly Labour-supporting, limiting the variance on the measure of hoped-for outcome. Additionally, it could be argued that even many of those voters not wholeheartedly welcoming the prospect of a Labour government may have felt some sense of relief that the rather tumultuous time (including five prime ministers) of the Tory government was over. This is reflected in the overwhelming loss of seats by the Conservative Party, not only to Labour, but also other parties.

We have therefore expanded the point that was already intimated in the Limitations section (the new text is blue:

“For instance, election outcomes in Germany and the UK were highly predictable compared to the US elections, and the sample was relatively homogenous in political beliefs, limiting variance on pre-post changes. This might for instance explain why the measure assessing which outcome participants hoped for did not predict increases in valence, possibly in combination with a

dissatisfaction with the government that transcended the ranks of Labour supporters.” (p. 28)